# Private Learning of Halfspaces: Simplifying the Construction & Reducing the Sample Complexity

**Haim Kaplan**
Tel Aviv University and Google Research
haimk@tau.ac.il

**Yishay Mansour**
Tel Aviv University and Google Research
mansour.yishay@gmail.com

**Uri Stemmer**
Ben-Gurion University and Google Research
u@uri.co.il

**Eliad Tsfadia**
Tel Aviv University and Google Research
eliadtsfadia@gmail.com

## Abstract

We present a differentially private learner for halfspaces over a finite grid $G$ in $\mathbb{R}^d$ with sample complexity $\approx d^{2.5} \cdot 2^{\log^* |G|}$, which improves the state-of-the-art result of [Beimel et al., COLT 2019] by a $d^2$ factor. The building block for our learner is a new differentially private algorithm for approximately solving the linear feasibility problem: Given a feasible collection of $m$ linear constraints of the form $Ax \geq b$, the task is to *privately* identify a solution $x$ that satisfies *most* of the constraints. Our algorithm is iterative, where each iteration determines the next coordinate of the constructed solution $x$.

## 1 Introduction

Machine learning is an extremely beneficial technology, helping us improve upon nearly all aspects of life. However, while the benefits of this technology are rather self-evident, it is not without risks. In particular, machine learning models are often trained on sensitive personal information, a fact which may pose serious privacy threats for the training data. These threats, together with the increasing awareness and demand for user privacy, motivated a long line of work focused on developing *private learning algorithms* that provide rigorous privacy guarantees for their training data.

We can think of a private learner as an algorithm that operates on a database containing *labeled* individual information, and outputs a hypothesis that predicts the labels of unseen individuals. For example, consider a medical database in which every row contains the medical history of one individual together with a yes/no label indicating whether this individual suffers from some disease. Given this database, a learning algorithm might try to predict whether a new patient suffers from this disease given her medical history. The privacy requirement is that, informally, the output of the learner (the chosen hypothesis) leaks very little information on any particular individual from the database. Formally,

**Definition 1.1** (Dwork et al. [2006]). Let $\mathcal{A}$ be a randomized algorithm that operates on databases. Algorithm $\mathcal{A}$ is $(\varepsilon, \delta)$-*differentially private* if for any two databases $\mathcal{S}, \mathcal{S}'$ that differ in one row, and any event $\mathcal{T}$, we have $\Pr[\mathcal{A}(\mathcal{S}) \in \mathcal{T}] \leq e^{\varepsilon} \cdot \Pr[\mathcal{A}(\mathcal{S}') \in \mathcal{T}] + \delta$. The definition is referred to as *pure* differential privacy when $\delta = 0$, and *approximate* differential privacy when $\delta > 0$.

When constructing private learners, there is a strong tension between the privacy requirement and the utility that can be achieved; one very important and natural measure for this tradeoff is the amount of data required to achieve both goals simultaneously, a.k.a. the *sample complexity*. This measure is crucial to the practice as it determines the amount of *individual data* that must be collected before starting the analysis in the first place.

Recall that the sample complexity of non-private learning is fully characterized by the VC dimension of the hypothesis class. For *pure*-private learners (i.e., learners that satisfy pure-differential privacy), there is an analogous characterizations in terms of a measure called *the representation dimension* [Beimel et al., 2013a]. However, the situation is far less understood for *approximate* private learning, and there is currently no tight characterization for the sample complexity of *approximate* private learners.[1]

In this work we investigate the sample complexity of private learning for one of the most basic and important learning tasks – learning halfspaces. We begin by surveying the existing results.

## 1.1 Existing Results

Recall that the VC dimension of the class of all halfspaces over $\mathbb{R}^d$ is $d$, and hence a sample of size $O(d)$ suffices to learn halfspaces non-privately (we omit throughout the introduction the dependency of the sample complexity in the accuracy, confidence, and privacy parameters). In contrast, it turns out that with differential privacy, learning halfspaces over $\mathbb{R}^d$ is *impossible*, even with approximate differential privacy, and even when $d = 1$ [Feldman and Xiao, 2015, Bun et al., 2015, Alon et al., 2019].

In more details, let $X \in \mathbb{N}$ be a discretization parameter, let $\mathcal{X} = \{x \in \mathbb{Z} : |x| \leq X\}$, and consider the task of learning halfspaces over the *finite* grid $\mathcal{X}^d \subseteq \mathbb{R}^d$. In other words, consider the task of learning halfspaces under the promise that the underlying distribution is supported on (a subset of) the finite grid $\mathcal{X}^d$. For pure-private learning, Feldman and Xiao [2015] showed a lower bound of $\Omega\left(d^2 \cdot \log X\right)$ on the sample complexity of this task. This lower bound is tight, as a pure-private learner with sample complexity $\Theta\left(d^2 \cdot \log X\right)$ can be obtained using the generic upper bound of Kasiviswanathan et al. [2011]. This should be contrasted with the non-private sample complexity, which is linear in $d$ and independent of $X$.

For the case of $d = 1$, Beimel et al. [2013b] showed that the lower bound of Feldman and Xiao [2015] can be circumvented by relaxing the privacy guarantees from pure to approximate differential privacy. Specifically, they presented an approximate-private learner for 1-dimensional halfspaces with sample complexity $2^{O(\log^* X)}$. The building block in their construction is a differentially private algorithm, called $\mathcal{A}_{\mathrm{RecConcave}}$, for approximately optimizing *quasi-concave* functions.[2]

Following the work of Beimel et al. [2013b], two additional algorithms for privately learning 1-dimensional halfspaces with sample complexity $2^{O(\log^* X)}$ were given by Bun et al. [2015] and by Bun et al. [2018]. Recently, an algorithm with sample complexity $\tilde{O}\left((\log^* X)^{1.5}\right)$ was given by Kaplan et al. [2020] (again for $d = 1$). In light of these positive results, it might be tempting to guess that the sample complexity of privately learning halfspaces can be made independent of the discretization parameter $X$. However, as Bun et al. [2015] and Alon et al. [2019] showed, this is not the case, and every approximate-private learner for 1-dimensional halfspaces over $\mathcal{X}$ must have sample complexity at least $\Omega(\log^* X)$. Observe that, in particular, this means that learning halfspaces over $\mathbb{R}$ is impossible with differential privacy (even for $d = 1$).

Recently, Beimel et al. [2019] presented an approximate-private learner for $d$-dimensional halfspaces (over $\mathcal{X}^d$) with sample complexity $\approx d^{4.5} \cdot 2^{O(\log^* X)}$. Their algorithm is based on a reduction to the task of privately finding a point in the convex hull of a given input dataset. Specifically, given a dataset $\mathcal{S}$ containing points from the finite grid $\mathcal{X}^d \subseteq \mathbb{R}^d$, consider the task of (privately) finding a point $y \in \mathbb{R}^d$ that belongs to the convex hull of the points in $\mathcal{S}$. Beimel et al. [2019] presented an iterative algorithm for this task that is based on the following paradigm: Suppose that we have identified values for the first $i - 1$ coordinates $x_1^*, \ldots, x_{i-1}^*$ for which we know that there exists a completion $\tilde{x}_i, \ldots, \tilde{x}_d$ such that $(x_1^*, \ldots, x_{i-1}^*, \tilde{x}_i, \ldots, \tilde{x}_d)$ belongs to the convex hull of the input points. Then, during the $i$th iteration of the algorithm, we aim to find the next coordinate $x_i^*$ such that $(x_1^*, \ldots, x_i^*)$ can be completed to a point in the convex hull. To that end, Beimel et al. [2019] formulated the task of identifying the next coordinate $x_i^*$ as a (1-dimensional) quasi-concave optimization problem,

and used algorithm $\mathcal{A}_{\text{RecConcave}}$ of Beimel et al. [2013b] for privately solving it. This strategy is useful because algorithm $\mathcal{A}_{\text{RecConcave}}$ is very efficient (in terms of sample complexity) in optimizing 1-dimensional quasi-concave functions (requires only $\approx 2^{O(\log^* X)}$ many samples). This paradigm (together with a reduction from privately learning halfspaces to privately finding a point in the convex hull) resulted in a private learner for halfspaces over $\mathcal{X}^d$ with sample complexity $\approx d^{4.5} \cdot 2^{O(\log^* X)}$.

For more related works, we refer to the supplementary material.

## 1.2 Our Results

In this work, we generalize the technique that Beimel et al. [2019] applied to the problem of finding a point in the convex hull, which we refer to as the "RecConcave paradigm", and reformulate it as a general method for privately optimizing high dimensional functions. As a result, we obtain a private PAC learner for halfspaces with an improved sample complexity of $\approx d^{2.5} \cdot 2^{O(\log^* X)}$.

**Theorem 1.2** (Learning Halfspaces, Informal)**.** *Let $\alpha, \beta, \varepsilon \leq 1$ and $\delta < 1/2$ and let $\mathcal{X} \subset \mathbb{R}$. There exists an $(\varepsilon, \delta)$-differentially private $(\alpha, \beta)$-PAC learner for halfspaces over examples from $\mathcal{X}^d$ with sample complexity $s = d^{2.5} \cdot 2^{O(\log^* X)} \cdot \frac{1}{\varepsilon\alpha} \cdot \text{polylog}\left(\frac{d}{\alpha\beta\varepsilon\delta}\right)$.*

To obtain Theorem 1.2, we show that the task of privately learning halfspaces reduces to the task of privately solving the linear feasibility problem (as defined below) with essentially the same parameters, and solve the linear feasibility problem using our generalized RecConcave paradigm.

**The Linear Feasibility Problem.** Let $\mathcal{X} = \{x \in \mathbb{Z} : |x| \leq X\}$ for some parameter $X \in \mathbb{N}$. In the linear feasibility problem, we are given a feasible collection of $m$ linear constraints over $d$ variables $x_1, \ldots, x_d$, and the goal is to find a solution in $\mathbb{R}^d$ that satisfies all constraints. Each constraint has the form $\sum_{i=1}^{d} a_i x_i \geq b$ for some $a_1, \ldots, a_d, b \in \mathcal{X}$.

Without privacy considerations, this well-known problem can be solved, e.g., using the Ellipsoid Method or the Interior Point Method. In the private version of this problem, we would like to come up with a solution to the system in a way that is insensitive to any (arbitrary) change of single constraint (in the sense of differential privacy, see Definition 1.1). It is easy to see that with differential privacy, one cannot hope for an exact solution to this problem (i.e., a solution that satisfies *all* constraints). This is because changing a single constraint, which has basically no effect on the outcome of a private algorithm, may completely change the feasibility area. Therefore, in the private version of this problem we only aim to satisfy *most* of the constraints. Specifically, we say that an algorithm $(\alpha, \beta)$-*solves* the $(X, d, m)$-linear feasibility problem, if for every feasible collection of $m$ linear constraints over $d$ variables with coefficients from $\mathcal{X}$, with probability $1 - \beta$ the algorithm finds a solution $\mathbf{x} = (x_1, \ldots, x_d)$ that satisfies at least $(1 - \alpha)m$ constraints.

**Question 1.3.** *What is the minimal number of constraints $m$, as a function of $X, d, \alpha, \beta, \varepsilon, \delta$, for which there exists an $(\varepsilon, \delta)$-differentially private algorithm that $(\alpha, \beta)$-solves the $(X, d, m)$-linear feasibility problem?*

Observe that this question is trivial without the privacy requirement (it can be solved easily when $m = 1$). However, the picture is quite different with differential privacy. In particular, all the lower bounds we mentioned before on the sample complexity of learning halfspaces yield lower bounds on the number of constraints $m$ needed to privately solve the linear feasibility problem. We prove the following theorem.

**Theorem 1.4** (Linear Feasibility Problem, Informal)**.** *Let $\alpha, \beta, \varepsilon \leq 1$ and $\delta < 1/2$ and let $X \in \mathbb{N}$. There exists an $(\varepsilon, \delta)$-differentially private algorithm that $(\alpha, \beta)$-solves the $(X, d, m)$-linear feasibility problem, for every $m \geq d^{2.5} \cdot 2^{O(\log^* X)} \cdot \frac{1}{\varepsilon\alpha} \cdot \text{polylog}\left(\frac{d}{\beta\delta}\right)$.*

**A Generalized RecConcave Paradigm.** Let $f(\mathcal{S}, \mathbf{x})$ be a low-sensitivity function that takes a database $\mathcal{S}$ and a high dimensional point $\mathbf{x}$, and returns a real number which is identified as the "score" of the point $\mathbf{x}$ w.r.t. the database $\mathcal{S}$.[3] Now suppose that, given an input database $\mathcal{S}$, we would like to (privately) identify a point $\mathbf{x}$ such that $f(\mathcal{S}, \mathbf{x})$ is approximately maximized.

**Example 1.5.** *To solve the linear feasibility problem we can define the function $f(\mathcal{S}, \mathbf{x})$ as the number of constraints in $\mathcal{S}$ that are satisfied by $\mathbf{x}$, a quantity which we denote by $\mathrm{depth}_\mathcal{S}(x_1, \ldots, x_d)$. Note that an approximate maximizer for this $f$ is a good solution to the linear feasibility problem, i.e., it satisfies most of the constraints.*

A naive attempt for using the RecConcave paradigm in order to privately maximize $f$ is to define the following function $Q$ (for every $i \in [d]$ and every fixing of $x_1^*, \ldots, x_{i-1}^*$).

$$Q_{x_1^*, \ldots, x_{i-1}^*}(x_i) = \max_{\tilde{x}_{i+1}, \ldots, \tilde{x}_d} \{ f\left( \mathcal{S}, x_1^*, \ldots, x_{i-1}^*, x_i, \tilde{x}_{i+1}, \ldots, \tilde{x}_d \right) \}.$$

Now, if it happens that $Q$ is quasi-concave, then one can apply $\mathcal{A}_{\mathrm{RecConcave}}$ coordinate by coordinate in order to privately find a solution $\mathbf{x}$ that approximately maximizes $f(\mathcal{S}, \mathbf{x})$. To see this, suppose that we find (using $\mathcal{A}_{\mathrm{RecConcave}}$) a value $x_1^*$ for the first coordinate that approximately maximizes $Q(\cdot)$. By the definition of $Q$, this guarantees that there exists a completion $(\tilde{x}_2, \ldots, \tilde{x}_d)$ such that $f(\mathcal{S}, x_1^*, \tilde{x}_2, \ldots, \tilde{x}_d)$ is almost as high as $\max_{\mathbf{x}} \{ f(\mathcal{S}, \mathbf{x}) \}$. Hence, by committing to $x_1^*$ we do not lose much in terms of the maximum attainable value of $f$. Similarly, in every iteration we identify a value for the next coordinate without losing too much in the maximum attainable value of $f$.

The problem is that, in general, the above function $Q$ is not necessarily quasi-concave. In particular, in the linear feasibility problem where $f(\mathcal{S}, \mathbf{x}) = \mathrm{depth}_\mathcal{S}(\mathbf{x})$ (i.e., the number of constraints in $\mathcal{S}$ that are satisfied by $\mathbf{x}$), the resulting function $Q$ is not quasi-concave.[4]

In order to overcome this issue, we present the following technique which we refer to as the generalized RecConcave Paradigm.[5] We define the "convexification" of a function $f$ to be the function $f_{\mathrm{Conv}}(\mathcal{S}, \mathbf{x})$ that outputs the maximal $y \in \mathbb{R}$ for which the point $\mathbf{x}$ is a convex combination of points $\mathbf{z} \in \mathbb{R}^d$ with $f(\mathcal{S}, \mathbf{z}) \geq y$. In other words, for any $y \in \mathbb{R}$, we consider the set $\mathcal{D}_\mathcal{S}(y) = \left\{ \mathbf{z} \in \mathbb{R}^d : f(\mathcal{S}, \mathbf{z}) \geq y \right\}$, and denote $\mathcal{C}_\mathcal{S}(y) = \mathrm{ConvexHull}(\mathcal{D}_\mathcal{S}(y))$. Then, $f_{\mathrm{Conv}}(\mathcal{S}, \mathbf{x}) := \max\{ y : \mathbf{x} \in \mathcal{C}_\mathcal{S}(y) \}$. We show that with this function $f_{\mathrm{Conv}}(\mathcal{S}, \mathbf{x})$, the resulting function

$$Q_{x_1^*, \ldots, x_{i-1}^*}(x_i) = \max_{\tilde{x}_{i+1}, \ldots, \tilde{x}_d} \{ f_{\mathrm{Conv}}\left( \mathcal{S}, x_1^*, \ldots, x_{i-1}^*, x_i, \tilde{x}_{i+1}, \ldots, \tilde{x}_d \right) \}$$

is indeed quasi-concave for any fixing of $x_1^*, \ldots, x_{i-1}^*$ (no matter how the function $f$ is defined). The function $f_{\mathrm{Conv}}$ can, therefore, be approximately maximized (privately) coordinate by coordinate using $\mathcal{A}_{\mathrm{RecConcave}}$. Furthermore, if $f$ has the property that points $\mathbf{x}$ with high $f_{\mathrm{Conv}}(\mathcal{S}, \mathbf{x})$ also have (somewhat) high $f(\mathcal{S}, \mathbf{x})$, then $f$ can be privately maximized (approximately) by maximizing the function $f_{\mathrm{Conv}}$. Going back to the linear feasibility problem, we denote by $\mathrm{cdepth}_\mathcal{S}(\mathbf{x}) = f_{\mathrm{Conv}}(\mathcal{S}, \mathbf{x})$ the convexification of the function $f(\mathcal{S}, \mathbf{x}) = \mathrm{depth}_\mathcal{S}(\mathbf{x})$. We then show that every point that has $\mathrm{cdepth} = (1 - \lambda) |\mathcal{S}|$ must have $\mathrm{depth} \geq (1 - (d+1)\lambda) |\mathcal{S}|$. Applying the aforementioned method on the function $\mathrm{depth}$ results in a differentially private algorithm for solving the $(X, d, m)$-linear feasibility problem whenever $m \gtrsim d^{2.5} \cdot 2^{O(\log^* X)}$.

## 2 Preliminaries

### 2.1 Preliminaries from Learning Theory

We next define the probably approximately correct (PAC) model of Valiant [1984]. A *concept class* $\mathcal{C}$ over $\mathcal{X}$ is a set of concepts (predicates) mapping $\mathcal{X}$ to $\{0, 1\}$. A learning algorithm is given examples sampled according to an unknown probability distribution $\mu$ over $\mathcal{X}$, and labeled according to an unknown *target* concept $c \in \mathcal{C}$. The *generalization error* of a hypothesis $h : X \to \{0, 1\}$ is defined as $\mathrm{error}_\mu(c, h) = \Pr_{x \sim \mu}[h(x) \neq c(x)]$. Algorithm $\mathcal{A}$ is an $(\alpha, \beta)$-*PAC learner* for $\mathcal{C}$ with sample complexity $m$ if for all concepts $c \in \mathcal{C}$, all distributions $\mu$ on $\mathcal{X}$, given an input of $m$ samples $\mathcal{S} = (z_1, \ldots, z_m)$, where $z_i = (x_i, c(x_i))$ and each $x_i$ is drawn i.i.d. from $\mu$, algorithm $\mathcal{A}$ outputs a hypothesis $h$ satisfying $\Pr[\mathrm{error}_\mu(c, h) \leq \alpha] \geq 1 - \beta$, where the probability is taken over the random choice of the examples in $\mathcal{S}$ according to $\mu$ and the random coins of the learner $\mathcal{A}$. For a labeled sample $\mathcal{S} = (x_i, y_i)_{i=1}^m$, the *empirical error* of $h$ is $\mathrm{error}_\mathcal{S}(h) = \frac{1}{m} |\{i : h(x_i) \neq y_i\}|$.

### 2.1.1 Private Learning

An Algorithm $\mathcal{A}$ is an $(\varepsilon, \delta)$-differential private $(\alpha, \beta)$-PAC for $\mathcal{C}$ with sample complexity $m$ if: (1) $\mathcal{A}$ is $(\varepsilon, \delta)$-differentially private (as in Definition 1.1), and (2) Algorithm $\mathcal{A}$ is an $(\alpha, \beta)$-*PAC learner* for $\mathcal{C}$ with sample complexity $m$.

## 2.2 A Private Algorithm for Optimizing Quasi-concave Functions – $\mathcal{A}_{\text{RecConcave}}$

We describe the properties of algorithm $\mathcal{A}_{\text{RecConcave}}$ of Beimel et al. [2013b]. This algorithm is given a quasi-concave function $Q$ (defined below) and a database $\mathcal{S}$ and privately finds a point $x$ such that $Q(\mathcal{S}, x)$ is close to its maximum provided that the maximum of $Q(\mathcal{S}, \cdot)$ is large enough.

**Definition 2.1.** A function $f$ is quasi-concave if $f(\ell) \geq \min\{f(i), f(j)\}$ for every $i < \ell < j$.

**Definition 2.2** (Sensitivity). The sensitivity of a function $f: \mathcal{X}^* \to \mathbb{R}$ is the smallest $k$ such that for every neighboring databases $\mathcal{S}, \mathcal{S}' \in \mathcal{X}^*$ (i.e., differ in exactly one entry), we have $|f(\mathcal{S}) - f(\mathcal{S}')| \leq k$. A function $g: \mathcal{X}^* \times \tilde{\mathcal{X}} \to \mathbb{R}$ is called a sensitivity-$k$ function if for every $x \in \tilde{\mathcal{X}}$, the function $g(\cdot, x)$ has sensitivity $\leq k$.

**Proposition 2.3** (Properties of Algorithm $\mathcal{A}_{\text{RecConcave}}$ [Beimel et al., 2013b]). *Let $Q : \mathcal{X}^* \times \tilde{\mathcal{X}} \to \mathbb{R}$ be a sensitivity-1 function. Denote $\tilde{X} = \left| \tilde{\mathcal{X}} \right|$ and let $\alpha \leq \frac{1}{2}$ and $\beta, \varepsilon, \delta, r$ be parameters. There exists an $(\varepsilon, \delta)$-differentially private algorithm, called $\mathcal{A}_{\text{RecConcave}}$, such that the following holds. If $\mathcal{A}_{\text{RecConcave}}$ is executed on a database $\mathcal{S} \in \mathcal{X}^*$ such that $Q(\mathcal{S}, \cdot)$ is quasi-concave and*

$$\max_{i \in \tilde{\mathcal{X}}}\{Q(\mathcal{S}, i)\} \geq r \geq 8^{\log^* \tilde{X}} \cdot \frac{12 \log^* \tilde{X}}{\alpha \varepsilon} \log \left( \frac{192 (\log^* \tilde{X})^2}{\beta \delta} \right). \tag{1}$$

*then with probability $1 - \beta$ the algorithm outputs an index $j$ s.t. $Q(\mathcal{S}, j) \geq (1 - \alpha)r$.*

Namely, when there exists a solution with a promised quality of at least $r$, then with probability $1 - \beta$ Algorithm $\mathcal{A}_{\text{RecConcave}}$ finds a solution with quality at least $(1 - \alpha)r$. We next give a short summary of how it works.

Beimel et al. [2013b] observed that a quasi-concave promise problem can be privately approximated using a solution to a smaller instance of a quasi-concave promise problem. Specifically, they showed that for any quasi-concave function $Q : \mathcal{X}^* \times \tilde{\mathcal{X}} \to \mathbb{R}$ with a (large enough) promise $r$, there exists a quasi-concave function $Q' : \mathcal{X}^* \times \tilde{\mathcal{X}}' \to \mathbb{R}$ with a promise $r' = \Omega(\alpha r)$ and with $\left| \tilde{\mathcal{X}}' \right| \approx \log \left| \tilde{\mathcal{X}} \right|$, such that the task of privately finding $j \in \tilde{\mathcal{X}}$ with $Q(\mathcal{S}, j) \geq (1 - \alpha)r$ is reduced to the task of privately finding $k \in \tilde{\mathcal{X}}'$ with $Q'(\mathcal{S}, k) \geq (1 - \alpha)r'$. This resulted in a recursive algorithm $\mathcal{A}_{\text{RecConcave}}$ for optimizing $Q$. For the sake of completeness, we give more details in the supplementary material.

## 2.3 Halfspaces and Convex Hull

We next define the geometric objects we use in this paper.

**Definition 2.4** (Halfspaces and Hyperplanes). For $\mathbf{a} = (a_1, \ldots, a_d) \in \mathbb{R}^d \setminus \{(0, \ldots, 0)\}$ and $w \in \mathbb{R}$, let the halfspace defined by $(\mathbf{a}, w)$ be $\text{hs}_{\mathbf{a}, w} := \{\mathbf{x} \in \mathbb{R}^d : \langle \mathbf{a}, \mathbf{x} \rangle \geq w\}$. For a domain $\mathcal{D} \subseteq \mathbb{R}^d$ define the concept class $\texttt{HALFSPACE}(\mathcal{D}) = \{c_{\mathbf{a}, w} : \mathcal{D} \mapsto \{-1, 1\}\}$, letting $c_{\mathbf{a}, w}$ be the function that on input $\mathbf{x} \in \mathcal{D}$ outputs 1 iff $\mathbf{x} \in \text{hs}_{\mathbf{a}, w}$.

**Definition 2.5** (Convex Hull). Let $\mathcal{P} \subseteq \mathbb{R}^d$ be a set of points. The convex hull of $\mathcal{P}$, denote by $\text{ConvexHull}(\mathcal{P})$, is the set of all points $\mathbf{x} \in \mathbb{R}^d$ that are convex combination of elements of $\mathcal{P}$. That is, $\mathbf{x} \in \text{ConvexHull}(\mathcal{P})$ iff there exists a finite subset $\mathcal{P}' \subseteq \mathcal{P}$ and numbers $\{\lambda_{\mathbf{y}}\}_{\mathbf{y} \in \mathcal{P}'}$ such that $\sum_{\mathbf{y} \in \mathcal{P}'} \lambda_{\mathbf{y}} = 1$ and $\sum_{\mathbf{y} \in \mathcal{P}'} \lambda_{\mathbf{y}} \mathbf{y} = \mathbf{x}$.

**Fact 2.6** (Caratheodory's theorem). Let $\mathcal{P} \subseteq \mathbb{R}^d$ be a set of points. Then any $\mathbf{x} \in \text{ConvexHull}(\mathcal{P})$ is a convex combination of at most $d + 1$ points in $\mathcal{P}$.

# 3 Optimizing High-Dimensional Functions

In this section we present our general method for privately optimizing high dimensional functions. In the following, let $\mathcal{X}$ be a domain and let $f: \mathcal{X}^* \times \mathbb{R}^d \to \mathbb{R}$ be a function that given a dataset

$S \in \mathcal{X}^*$, we would like to approximately maximize $f(S, \cdot)$. Formally, given $\alpha, \beta, \epsilon, \delta \in (0, 1)$, our goal is to design an $(\varepsilon, \delta)$-differential private algorithm that with probability $1 - \beta$ finds $\mathbf{x}^* \in \mathbb{R}^d$ with $f(S, \mathbf{x}^*) \geq (1 - \alpha)M_S$ for $M_S := \max_{\mathbf{x}} f(S, \mathbf{x})$. We do so by optimizing a different (but related) function $f_{\mathrm{Conv}}$, which we call the "convexification" of $f$.

**Definition 3.1** (The convexification of $f$). For $S \in \mathcal{X}^*$ and $y \in \mathbb{R}$, let $\mathcal{D}_S(y) := \{\mathbf{z} \in \mathbb{R}^d : f(S, \mathbf{z}) \geq y\}$ and $\mathcal{C}_S(y) := \mathrm{ConvexHull}(\mathcal{D}_S(y))$. We define the convexification of $f$ as the function $f_{\mathrm{Conv}} : \mathcal{X}^* \times \mathbb{R}^d \to \mathbb{R}$ defined by $f_{\mathrm{Conv}}(S, \mathbf{x}) := \max\{y \in \mathbb{R} : \mathbf{x} \in \mathcal{C}_S(y)\}$.

Namely, $f_{\mathrm{Conv}}(S, \mathbf{x}) = y$ if and only if $y$ is the maximal value such that $\mathbf{x}$ is a convex combination of points $\mathbf{z}$ with $f(S, \mathbf{z}) \geq y$. Note that by definition it is clear that $f(S, \mathbf{x}) \leq f_{\mathrm{Conv}}(S, \mathbf{x})$ for any $(S, \mathbf{x}) \in \mathcal{X}^* \times \mathbb{R}^d$. Yet, observe that $\max_{\mathbf{x}} f_{\mathrm{Conv}}(S, \mathbf{x}) = M_S$.

In the following, assume that points with high value of $f_{\mathrm{Conv}}$ also have somewhat high value of $f$. Formally, assume there exists $\Delta \geq 1$ that satisfies the following requirement:

**Requirement 3.2.** $\forall (S, \mathbf{x}) \in \mathcal{X}^* \times \mathbb{R}^d : f(S, \mathbf{x}) \geq \Delta \cdot f_{\mathrm{Conv}}(S, \mathbf{x}) - (\Delta - 1) \cdot M_S$

Requirement 3.2 can be interpreted as follows: For any $(S, \mathbf{x}) \in \mathcal{X}^* \times \mathbb{R}^d$, if $f_{\mathrm{Conv}}(S, \mathbf{x}) = (1 - \lambda)M_S$, then $f(S, \mathbf{x}) \geq (1 - \lambda\Delta)M_S$. This reduces the task of finding a point $\mathbf{x}^*$ with $f(S, \mathbf{x}^*) \geq (1 - \alpha)M_S$ to the task of finding a point $\mathbf{x}^*$ with $f_{\mathrm{Conv}}(S, \mathbf{x}^*) \geq (1 - \alpha/\Delta)M_S$.

Following the above assumption, the idea of our algorithm is to find a point $\mathbf{x}^* = (x_1^*, \ldots, x_d^*)$ with large $f_{\mathrm{Conv}}$ coordinate after coordinate: we use $\mathcal{A}_{\mathrm{RecConcave}}$ to find a value $x_1^*$ that can be extended by some $\tilde{x}_2, \ldots, \tilde{x}_d$ so that $f_{\mathrm{Conv}}(x_1^*, \tilde{x}_2 \ldots, \tilde{x}_d)$ is close to $M_S$, then we find a value $x_2^*$ so that there is a point $(x_1^*, x_2^*, \tilde{x}_3 \ldots, \tilde{x}_d)$ whose $f_{\mathrm{Conv}}$ is close to $M_S$, and so forth until we find all coordinates. The parameters in $\mathcal{A}_{\mathrm{RecConcave}}$ are set such that in each step we lose at most $\alpha M_S / (d\Delta)$ from the value of $f_{\mathrm{Conv}}$, resulting in a point $(x_1^*, \ldots, x_d^*)$ whose $f_{\mathrm{Conv}}$ is at least $(1 - \alpha/\Delta)M_S$.

## 3.1 Defining a Quasi-Concave Function with Small Sensitivity

To apply the above approach, we need to prove that the functions considered in the algorithm $\mathcal{A}_{\mathrm{RecConcave}}$ are quasi-concave and have small sensitivity of the dataset $S$.

**Definition 3.3.** For $1 \leq i \leq d$ and $x_1^*, \ldots, x_{i-1}^* \in \mathbb{R}$, define

$$Q_{x_1^*, \ldots, x_{i-1}^*}(S, x_i) := \max_{\tilde{x}_{i+1}, \ldots, \tilde{x}_d \in \mathbb{R}} f_{\mathrm{Conv}}(S, x_1^*, \ldots, x_{i-1}^*, x_i, \tilde{x}_{i+1}, \ldots, \tilde{x}_d).$$

We first prove that the function $Q_{x_1^*, \ldots, x_{i-1}^*}(S, \cdot)$ is quasi-concave .

**Claim 3.4.** For every $i \in [d]$ and $x_1^*, \ldots, x_{i-1}^* \in \mathbb{R}$, the function $Q_{x_1^*, \ldots, x_{i-1}^*}(S, \cdot)$ is quasi-concave.

*Proof.* Fix $i \in [d]$ and $x_1^*, \ldots, x_{i-1}^* \in R$, and fix values $x_i, x_i', x_i'' \in \mathbb{R}$ such that $x_i' \leq x_i \leq x_i''$, and let $y := \min\{Q_{x_1^*, \ldots, x_{i-1}^*}(S, x_i'), Q_{x_1^*, \ldots, x_{i-1}^*}(S, x_i'')\}$. By definition, $\exists x_{i+1}', \ldots, x_d', x_{i+1}'', \ldots, x_d'' \in \mathbb{R}$ such that both points $\mathbf{x}' = (x_1^*, \ldots, x_{i-1}^*, x_i', x_{i+1}', \ldots, x_d')$ and $\mathbf{x}'' = (x_1^*, \ldots, x_{i-1}^*, x_i'', x_{i+1}'', \ldots, x_d'')$ belong to $\mathcal{C}_S(y)$. In the following, let $p \in [0, 1]$ be the value such that $x_i = px_i' + (1-p)x_i''$, and let $\mathbf{x} = (x_1^*, \ldots, x_{i-1}^*, x_i, x_{i+1}, \ldots, x_d)$ where $x_j = px_j' + (1-p)x_j''$ for $j \in \{i+1, \ldots, d\}$. Since $\mathbf{x}$ lies on the line segment between $\mathbf{x}'$ and $\mathbf{x}''$, it holds that $\mathbf{x} \in \mathcal{C}_S(y)$ (recall that $\mathcal{C}_S(y)$ is a convex set). Therefore, we conclude that $Q_{x_1^*, \ldots, x_{i-1}^*}(x_i) \geq y$, as required. $\qquad\square$

We next prove that $Q_{x_1^*, \ldots, x_{i-1}^*}(\cdot, x_i)$ has low sensitivity.

**Claim 3.5.** Assume that $f$ is a sensitivity-$k$ function. Then for all $i \in [d]$ and $x_1^*, \ldots, x_{i-1}^* \in R$, $Q_{x_1^*, \ldots, x_{i-1}^*}$ is a sensitivity-$k$ function.

*Proof.* Fix two neighboring datasets $S, S' \in \mathcal{X}^*$. By assumption, it holds that $f(S, \mathbf{x}) \geq f(S', \mathbf{x}) - k$ for every $\mathbf{x} \in \mathbb{R}^d$. This yields that $\mathcal{C}_{S'}(y) \subseteq \mathcal{C}_S(y - k)$ for every $y \in \mathbb{R}$. Hence, we deduce by the definition of $Q_{x_1^*, \ldots, x_{i-1}^*}$ that $Q_{x_1^*, \ldots, x_{i-1}^*}(S, x_i) \geq Q_{x_1^*, \ldots, x_{i-1}^*}(S', x_i) - k$ for every $x_i \in \mathbb{R}$. $\qquad\square$

In order to apply algorithm $\mathcal{A}_{\mathrm{RecConcave}}$, for every $1 \leq i \leq d$ it is required to determine a finite domain $\tilde{\mathcal{X}}_i = \tilde{\mathcal{X}}_i(x_1^*, \ldots, x_{i-1}^*)$ which contains a value $x_i^*$ that reaches the maximum of

$Q_{x_1^*,\ldots,x_{i-1}^*}(\mathcal{S},\cdot)$ under $\mathbb{R}$.[6] Namely, we need to determined an iterative sequence of domains $\left\{\tilde{\mathcal{X}}_i(\cdot)\right\}_{i=1}^d$ that satisfies the following requirement:

**Requirement 3.6.** *For every $\mathcal{S} \in \mathcal{X}^*$ and every $x_1^*, \ldots, x_{i-1}^* \in \mathbb{R}$, it holds that*

$$\exists x_i \in \tilde{\mathcal{X}}_i : \ Q_{x_1^*,\ldots,x_{i-1}^*}(\mathcal{S}, x_i) = \max_{\tilde{x}_i \in \mathbb{R}} Q_{x_1^*,\ldots,x_{i-1}^*}(\mathcal{S}, \tilde{x}_i).$$

### 3.2 The Algorithm

In Figure 1, we present an $(\varepsilon, \delta)$-differentially private algorithm $\mathcal{A}_{\text{OptimizeHighDimFunc}}$ that finds with probability at least $1 - \beta$ a point $\mathbf{x}^* \in \mathbb{R}^d$ with $f(\mathcal{S}, \mathbf{x}^*) \geq (1 - \alpha)M_{\mathcal{S}}$.

---

**Algorithm $\mathcal{A}_{\text{OptimizeHighDimFunc}}$**

(i) Let $\alpha, \beta, \varepsilon, \delta \in (0,1)$ be the utility/privacy parameters, let $\mathcal{S} \in \mathcal{X}^*$ be an input dataset, let $\left\{\tilde{\mathcal{X}}_i(\cdot)\right\}_{i=1}^d$ be an iterative sequence of finite domains, and let $\Delta \geq 1$.

(ii) For $i = 1$ to $d$ do:

    (a) Let $Q_{x_1^*,\ldots,x_{i-1}^*}$ be the function from Definition 3.3.

    (b) Let $\tilde{\mathcal{X}}_i = \tilde{\mathcal{X}}_i(x_1^*, \ldots, x_{i-1}^*)$.

    (c) Execute $\mathcal{A}_{\text{RecConcave}}$ with the function $Q_{x_1^*,\ldots,x_{i-1}^*}$, domain $\tilde{\mathcal{X}}_i$, and parameters:
$$r = (1 - \tfrac{\alpha}{2d\Delta})^{i-1}M_{\mathcal{S}}, \tilde{\alpha} = \tfrac{\alpha}{2d\Delta}, \tilde{\beta} = \tfrac{\beta}{d}, \tilde{\varepsilon} = \tfrac{\varepsilon}{2\sqrt{2d\ln(2/\delta)}}, \tilde{\delta} = \tfrac{\delta}{2d}.$$
    Let $x_i^*$ be its output.

(iii) Return $\mathbf{x}^* = (x_1^*, \ldots, x_d^*)$.

---

Figure 1: Algorithm for finding a point $\mathbf{x}^* \in \mathbb{R}^d$ with $f(\mathcal{S}, \mathbf{x}^*) \geq (1 - \alpha)M_{\mathcal{S}}$.

The following theorem summarizes the properties of $\mathcal{A}_{\text{OptimizeHighDimFunc}}$.

**Theorem 3.7.** *Let $\mathcal{X}$ be a domain and $f \colon \mathcal{X}^* \times \mathbb{R}^d \to \mathbb{R}$ be a sensitivity-1 function. Let $\Delta \geq 1$ be a value that satisfies Requirement 3.2, and let $\left\{\tilde{\mathcal{X}}_i(\cdot)\right\}_{i=1}^d$ be an iterative sequence of finite domains that satisfies Requirement 3.6 (all with respect to $f$). In addition, let $\alpha, \beta, \varepsilon \leq 1$, $\delta < 1/2$, and let $\mathcal{S} \in \mathcal{X}^*$ be a dataset with $M_{\mathcal{S}} := \max_{\mathbf{x} \in \mathbb{R}^d} f(\mathcal{S}, \mathbf{x}) \geq \Omega\left(\Delta \cdot d^{1.5} \cdot 2^{O(\log^* \tilde{X})} \cdot \frac{\log^{1.5}\left(\frac{1}{\delta}\right)\log\left(\frac{d}{\beta}\right)}{\varepsilon\alpha}\right)$, where $\tilde{X} := \max_{i, x_1^*, \ldots, x_{i-1}^*}\left|\tilde{\mathcal{X}}_i(x_1^*, \ldots, x_{i-1}^*)\right|$. Then, $\mathcal{A}_{\text{OptimizeHighDimFunc}}$ is an $(\varepsilon, \delta)$-differentially private algorithm that with probability $1 - \beta$ returns a point $\mathbf{x}^* \in \mathbb{R}^d$ with $f(\mathcal{S}, \mathbf{x}^*) \geq (1 - \alpha)M_{\mathcal{S}}$.*

By Claims 3.4 and 3.5, the proof of Theorem 3.7 follows similarly to Theorem 20 of Beimel et al. [2019] using the properties of $\mathcal{A}_{\text{RecConcave}}$. See the full version for more details.

## 4 The Linear Feasibility Problem

In this section we show how the method from Section 3 can be used for privately approximating the linear feasibility problem. In this problem, we are given a finite grid $\mathcal{X} = [[\pm X]] := \{x \in \mathbb{Z} \colon |x| \leq X\}$ for some $X \in \mathbb{N}$ and a dataset $\mathcal{S} \in (\mathcal{X}^d \times \mathcal{X})^*$ such that each $(\mathbf{a}, w) \in \mathcal{S}$ represents the linear constraint $\langle \mathbf{a}, \mathbf{x} \rangle \geq w$ which defines the halfspace $\text{hs}_{\mathbf{a},w}$ in $\mathbb{R}^d$. In the following, we let $\text{depth}_{\mathcal{S}}(\mathbf{x}) := |\{(\mathbf{a}, w) \in \mathcal{S} \colon \mathbf{x} \in \text{hs}_{\mathbf{a},w}\}|$ (that is, the number of halfspaces in $\mathcal{S}$ that contain the point $\mathbf{x}$). Our goal is to describe, given $\alpha, \beta, \varepsilon, \delta \in (0,1)$, an $(\varepsilon, \delta)$-differential private algorithm that

satisfies the following utility guarantee: Given a realizable dataset of halfspaces (i.e., there exists a point $\mathbf{x} \in \mathbb{R}^d$ with $\mathrm{depth}_{\mathcal{S}}(\mathbf{x}) = |\mathcal{S}|$), then with probability $1 - \beta$ the algorithm should output a point $\mathbf{x}^*$ with $\mathrm{depth}_{\mathcal{S}}(\mathbf{x}^*) \geq (1 - \alpha) |\mathcal{S}|$.

In the following, let cdepth be the convexification of the function depth (according to Definition 3.1). That is, $\mathrm{cdepth}_{\mathcal{S}}(\mathbf{x}) = f_{\mathrm{Conv}}(\mathcal{S}, \mathbf{x})$ for the function $f(\mathcal{S}, \mathbf{x}) = \mathrm{depth}_{\mathcal{S}}(\mathbf{x})$. As a first step towards applying Theorem 3.7 for maximizing depth, we need to determine a value $\Delta \geq 1$ that satisfies Requirement 3.2. Namely, we need to lower bound $\mathrm{depth}_{\mathcal{S}}(\mathbf{x})$ in terms of $\mathrm{cdepth}_{\mathcal{S}}(\mathbf{x})$ and $M_{\mathcal{S}} = |\mathcal{S}|$. The following claim proves that $\Delta = d + 1$ satisfies Requirement 3.2 for the function depth.

**Claim 4.1.** *For any $\mathcal{S} \in (\mathbb{R}^d \times \mathbb{R})^*$ and $\mathbf{x} \in \mathbb{R}^d$, it holds that*

$$\mathrm{depth}_{\mathcal{S}}(\mathbf{x}) \geq (d + 1) \cdot \mathrm{cdepth}_{\mathcal{S}}(\mathbf{x}) - d |\mathcal{S}| .$$

*Proof.* Fix $\mathcal{S} \in (\mathbb{R}^d \times \mathbb{R})^*$ and $\mathbf{x} \in \mathbb{R}^d$, and let $k = \mathrm{cdepth}(\mathbf{x})$. By definition it holds that $\mathbf{x} \in \mathrm{ConvexHull}(\mathcal{D}_{\mathcal{S}}(k))$ for $\mathcal{D}_{\mathcal{S}}(k) = \{\mathbf{x}' \colon \mathrm{depth}_{\mathcal{S}}(\mathbf{x}') \geq k\}$. Therefore, by Caratheodory's theorem (Fact 2.6) it holds that $\mathbf{x}$ is a convex combination of at most $d + 1$ points $\mathbf{x}_1, \ldots, \mathbf{x}_{d+1} \in \mathcal{D}_{\mathcal{S}}(k)$. In the following, for $\mathbf{x}' \in \mathbb{R}^d$ let $\mathcal{T}_{\mathbf{x}'} := \{(\mathbf{a}, w) \in \mathcal{S} \colon \mathbf{x}' \notin \mathrm{hs}_{\mathbf{a},w}\}$ and observe that $\mathrm{depth}_{\mathcal{S}}(\mathbf{x}') = |\mathcal{S}| - |\mathcal{T}_{\mathbf{x}'}|$. Therefore, because for all $i \in [d + 1]$ we have $\mathrm{depth}(\mathbf{x}_i) \geq k$, it holds that $|\mathcal{T}_{\mathbf{x}_i}| \leq |\mathcal{S}| - k$. Furthermore, note that $\mathcal{T}_{\mathbf{x}} \subseteq \bigcup_{i=1}^{d} \mathcal{T}_{\mathbf{x}_i}$ (holds since each halfspace that contains a set of points also contains any convex combination of them). We conclude that $\mathrm{depth}_{\mathcal{S}}(\mathbf{x}) \geq |\mathcal{S}| - \sum_{i=1}^{d+1} |\mathcal{T}_{\mathbf{x}_i}| \geq |\mathcal{S}| - (d + 1)(|\mathcal{S}| - k) = (d + 1)k - d |\mathcal{S}|$. $\square$

The second step towards applying Theorem 3.7 is to determine an iterative sequence of finite domains $\left\{ \tilde{\mathcal{X}}_i(\cdot) \right\}_{i=1}^{d}$ that satisfies Requirement 3.6. Namely, our goal is to determine a finite grid $\tilde{\mathcal{X}}_i = \tilde{\mathcal{X}}_i(x_1^*, \ldots, x_{i-1}^*)$ such that there exists $x_i^* \in \tilde{\mathcal{X}}_i$ that reaches the maximum of $Q_{x_1^*, \ldots, x_{i-1}^*}(\mathcal{S}, \cdot)$ under $\mathbb{R}$, where $Q_{x_1^*, \ldots, x_{i-1}^*}(\mathcal{S}, x_i) := \max_{\tilde{x}_{i+1}, \ldots, \tilde{x}_d \in \mathbb{R}} \mathrm{cdepth}_{\mathcal{S}}(x_1^*, \ldots, x_{i-1}^*, \tilde{x}_i, \ldots, \tilde{x}_d)$. The following claim specifies these grids. We prove it in the supplementary material.

**Claim 4.2.** *There exists a sequence $\left\{ \tilde{\mathcal{X}}_i(\cdot) \right\}_{i=1}^{d}$ with $\max_{i, x_1^*, \ldots, x_{i-1}^*} \left| \tilde{\mathcal{X}}_i(x_1^*, \ldots, x_{i-1}^*) \right| \leq (dX)^{2d^2}$ such that for all $i \in [d]$ and $x_1^*, \ldots, x_{i-1}^* \in \mathbb{R}$, $\exists x_i^* \in \mathbb{R}$ with $Q_{x_1^*, \ldots, x_{i-1}^*}(\mathcal{S}, x_i^*) = \max_{x_i \in \mathbb{R}} \left\{ Q_{x_1^*, \ldots, x_{i-1}^*}(\mathcal{S}, x_i) \right\}$.*

Using Claims 4.1 and 4.2 along with the fact that depth is a sensitivity-1 function, we now can apply Theorem 3.7 to obtain our main result about the linear feasibility problem. See the supplementary material for the running time analysis.

**Theorem 4.3** (Restatement of Theorem 1.4). *Let $\alpha, \beta, \varepsilon \leq 1$, $\delta < 1/2$, $X \in \mathbb{N}$, $\mathcal{X} = [[\pm X]]$ and let $\mathcal{S} \in \left( \mathcal{X}^d \times \mathcal{X} \right)^*$ be a realizable dataset with $|\mathcal{S}| = O\left( d^{2.5} \cdot 2^{O(\log^* X + \log^* d)} \frac{\log^{1.5}\left(\frac{1}{\delta}\right) \log\left(\frac{d}{\beta}\right)}{\varepsilon \alpha} \right)$. Let $\mathcal{A}_{\mathrm{FindDeepPoint}}$ be the algorithm that executes $\mathcal{A}_{\mathrm{OptimizeHighDimFunc}}$ on the function $f(\mathcal{S}, \cdot) = \mathrm{depth}_{\mathcal{S}}(\cdot)$, with parameters $\alpha, \beta, \varepsilon, \delta$, $\Delta = d + 1$ and the sequence of grids $\left\{ \tilde{\mathcal{X}}_i(\cdot) \right\}_{i=1}^{d}$ from Claim 4.2. Then, $\mathcal{A}_{\mathrm{FindDeepPoint}}$ is an $(\varepsilon, \delta)$-differentially private algorithm that with probability $1 - \beta$ returns a point $\mathbf{x}^* \in \mathbb{R}^d$ with $\mathrm{depth}(\mathbf{x}^*) \geq (1 - \alpha) |\mathcal{S}|$. The algorithm runs in time $T = \mathrm{poly}(d) \cdot |\mathcal{S}| \cdot \left( |\mathcal{S}|^d \cdot \log X + \mathrm{polylog}(1/\alpha, 1/\beta, 1/\varepsilon, 1/\delta, X) \right)$.*

## 5 Learning Halfspaces

In this section we describe our private empirical risk minimization (ERM) learner of halfspaces, and at the end we state our (almost) immediate corollary about private PAC learning.

In the considered problem, we are given a finite grid $\mathcal{X} = [[\pm X]] := \{x \in \mathbb{Z} \colon |x| \leq X\}$ for some $X \in \mathbb{N}$ and a dataset of labeled points $\mathcal{S} \in (\mathcal{X}^d \times \{-1, 1\})^*$. We say that $\mathcal{S}$ is a realizable dataset of points if there exists $(\mathbf{a}, w) \in \mathbb{R}^d \times \mathbb{R}$ with $\mathrm{error}_{\mathcal{S}}(c_{\mathbf{a},w}) := |\{(\mathbf{x}, y) \in \mathcal{S} \colon c_{\mathbf{a},w}(\mathbf{x}) \neq y\}| / |\mathcal{S}| = 0$, where $c_{\mathbf{a},w} \colon \mathcal{X}^d \mapsto \{-1, 1\}$ is the concept function that outputs 1 iff $\mathbf{x} \in \mathrm{hs}_{\mathbf{a},w}$. Our goal is to describe, given $\alpha, \beta, \varepsilon, \delta \in (0, 1)$, an $(\varepsilon, \delta)$-differential private algorithm that satisfies the following

utility guarantee: Given a realizable dataset of points $\mathcal{S}$, the algorithm should output with probability $1 - \beta$ a pair $(\mathbf{a}^*, w^*)$ with $\mathrm{error}_{\mathcal{S}}(c_{\mathbf{a}^*, w^*}) \leq \alpha$.

## 5.1 A Reduction to the Linear Feasibility Problem

We reduce the problem of learning a halfspace to the linear feasibility problem by using geometric duality between points and halfspaces. Formally, we translate a halfspace $\mathrm{hs}_{\mathbf{a}, w}$ to the point $(\mathbf{a}, w) \in \mathbb{R}^{d+1}$, and translate a labeled point $(\mathbf{x}, y) \in \mathcal{S}$ to the $(d+1)$-dimensional halfspace $\mathrm{hs}_{(y \cdot \mathbf{x}, -y), 0}$ which equals to $\{(\mathbf{a}, w) \in \mathbb{R}^{d+1} \colon \langle \mathbf{a}, \mathbf{x} \rangle \geq w\}$ if $y = 1$, and to $\{(\mathbf{a}, w) \in \mathbb{R}^{d+1} \colon \langle \mathbf{a}, \mathbf{x} \rangle \leq w\}$ if $y = -1$. By definition, for any realizable dataset of points $\mathcal{S}$, the multiset $\mathcal{S}' = \{((y \cdot \mathbf{x}, -y), 0) \colon (\mathbf{x}, y) \in \mathcal{S}\}$ is a realizable dataset of halfspaces. Therefore, by applying $\mathcal{A}_{\mathrm{FindDeepPoint}}$ on $\mathcal{S}'$ we obtain a deep point $(\mathbf{a}^*, w^*) \in \mathbb{R}^{d+1}$ for $\mathcal{S}'$, meaning that $\langle \mathbf{a}^*, y \cdot \mathbf{x} \rangle \geq y \cdot w^*$ for most of the $(\mathbf{x}, y) \in \mathcal{S}$, which is (almost) what we need. The problem is that the pairs $(\mathbf{x}, -1) \in \mathcal{S}$ with $\langle \mathbf{a}^*, \mathbf{x} \rangle = w^*$ do not count as points in $\mathrm{hs}_{\mathbf{a}^*, w^*}$ while they do count for the depth of $(\mathbf{a}^*, w^*)$ in $\mathcal{S}'$. Yet, assuming the points in $\mathcal{S}$ are in general position (an assumption that can be eliminated), then there can be at most $d$ such points. Since $d < \alpha |\mathcal{S}| / 2$ in our case, then $\mathrm{error}_{\mathcal{S}}(c_{\mathbf{a}, w})$ remains small.

Following the above discussion, we now state our result in the general position case.

**Theorem 5.1.** *Let $\alpha, \beta, \varepsilon \leq 1$, $\delta < 1/2$, $X \in \mathbb{N}$, $\mathcal{X} = [[\pm X]]$ and let $\mathcal{S} \in \left( \mathcal{X}^d \times \{-1, 1\} \right)^*$ be a realizable dataset of points with $|\mathcal{S}| = O\left( d^{2.5} \cdot 2^{O(\log^* X + \log^* d)} \frac{\log^{1.5}\left(\frac{1}{\delta}\right) \log\left(\frac{d}{\beta}\right)}{\varepsilon \alpha} \right)$.*

*Let $\mathcal{A}_{\mathrm{LearnHalfSpace}}$ be the algorithm the executes $\mathcal{A}_{\mathrm{FindDeepPoint}}$ on the multiset $\mathcal{S}' := \{((y \cdot \mathbf{x}, -y), 0) \colon (\mathbf{x}, y) \in \mathcal{S}\}$ and parameters $\alpha/2, \beta, \varepsilon, \delta$, and outputs the resulting point $(\mathbf{a}^*, w^*) \in \mathbb{R}^{d+1}$. Then $\mathcal{A}_{\mathrm{LearnHalfSpace}}$ is $(\varepsilon, \delta)$-differentially private. Moreover, assuming the points in $\mathcal{S}$ are in general position,[7] then with probability $1 - \beta$ it holds that $\mathrm{error}_{\mathcal{S}}(c_{\mathbf{a}^*, w^*}) \leq \alpha$.*

In the supplementary material we show how to remove the assumption that the points in $\mathcal{S}$ are in general position. Hence, since the VC dimension of $\mathtt{HALFSPACE}(\mathbb{R}^d)$ is only $d + 1$, we immediately obtain a private PAC learner from our private ERM learner, which is the main result of this paper.

**Theorem 5.2** (Restatement of Theorem 1.2). *Let $\alpha, \beta, \varepsilon, \delta, X, \mathcal{X}$ as in Theorem 5.1. Then there exists an $(\varepsilon, \delta)$-differentially private $(\alpha, \beta)$-PAC learner for the class $\mathtt{HALFSPACE}(\mathcal{X}^d)$ with sample complexity $s = O\left( d^{2.5} \cdot 2^{O\left( \log^* X + \log^* d + \log^*\left( \frac{1}{\alpha \beta \varepsilon \delta} \right) \right)} \cdot \frac{\log^{1.5}\left( \frac{1}{\delta} \right) \log\left( \frac{d}{\alpha \beta} \right)}{\varepsilon \alpha} \right)$.*

## 6 Open Questions

It still remains open what is the minimal sample complexity that is required for learning halfspaces with an (approximate) differential privacy. Our work provides a new upper bound of $\approx d^{2.5} \cdot 2^{O(\log^* X)}$ which improves the state-of-the-art result of Beimel et al. [2019] by a $d^2$ factor, and improves the generic upper bound of Kasiviswanathan et al. [2011] whenever (roughly) $d < \log^2 X$.[8] Yet, there is still a gap from the best known lower bound of $\Omega(d \cdot \log^* X)$ for proper learning (Bun et al. [2015]) and $\Omega(d + \log^* X)$ for improper learning. In particular, it still remains open whether we can avoid the exponential dependency in $\log^* X$ for $d > 1$. One option for answering it is by finding a different 1-dimensional quasi-concave optimization that only requires polynomial dependency in $\log^* X$, since RecConcave, the optimization that we are using, requires exponential dependency. Indeed, a recent work of Kaplan et al. [2020] shows an (almost) linear dependency in $\log^* X$ for 1-dimensional thresholds, which is a special case of a quasi-concave optimization, and it still remains open whether this result can be extended to the quasi-concave optimization case.

## Broader Impact

In this work we develop algorithms that maintain the differential privacy of the samples. When the samples represent individuals, our work helps to maintain the privacy of those individuals.

## Acknowledgments and Disclosure of Funding

Haim Kaplan is partially supported by Israel Science Foundation (grant 1595/19), German-Israeli Foundation (grant 1367/2017), and the Blavatnik Family Foundation.

Yishay Mansour has received funding from the European Research Council (ERC) under the European Union's Horizon 2020 research and innovation program (grant agreement No. 882396), and by the Israel Science Foundation (grant number 993/17).

Uri Stemmer is supported in part by the Israel Science Foundation (grant 1871/19), and by the Cyber Security Research Center at Ben-Gurion University of the Negev.

## Footnotes

[1]We remark that there is a loose characterization for private learning in terms of the *Littlestone dimension* [Alon et al., 2019, Bun et al., 2020]. Specifically, these results state that the sample complexity of privately learning a class $C$ is somewhere between $\Omega(\log^* L)$ and $2^{O(L)}$, where $L$ is the Littlestone dimension of $C$. In our context, for learning halfspaces, these results do not provide meaningful bounds on the sample complexity.

[2]A function $Q$ is *quasi-concave* if for any $x' \leq x \leq x''$ it holds that $Q(x) \geq \min\{Q(x'), Q(x'')\}$.

[3]The *sensitivity* of the function $f$ is the maximal difference by which the value of $f(\mathcal{S}, \mathbf{x})$ can change when modifying one element of the database $\mathcal{S}$. See Section 2 for a formal definition.

[4]For instance, consider the 2-dimensional constraints $x_2 \geq x_1$ and $x_2 \leq -x_1$. Then under the fixing $x_1^* = 1$, the depth of $x_2 = 0$ is 0 while the depth of $x_2 \in \{-1, 1\}$ is 1, yielding that $Q_{x_1^*}(0) < \min \left\{ Q_{x_1^*}(-1), Q_{x_1^*}(1) \right\}$, and so $Q_{x_1^*}$ is not quasi-concave.

[5]We remark that the presentation here is oversimplified, and hides many of the challenges that arise in the actual analysis.

[6]We remark that this step might be involved for some $d$-dimensional functions, but is inherent for privately optimizing them (at least if the optimization is done coordinate by coordinate). Yet, once we determine such domains with some finite bound $\tilde{X}$ on their sizes, it usually not blows up the resulting sample complexity of our algorithm since it only depends on $2^{O(\log^* \tilde{X})}$ (see Theorem 3.7).

[7] For us, a set of points in $\mathbb{R}^d$ are in general position if there are no $d+1$ points that lie on the same hyperplane.

[8] We remark that even when $d > \log^2 X$, we offer significant improvements over the generic learner in terms of runtime. In particular, our algorithm runs in time (roughly) $n^d$, where $n$ is the number of samples, while the generic learner has a runtime of at least $X^{d^2}$.

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
