[Supplementary Material]

# Private Learning of Halfspaces: Simplifying the Construction and Reducing the Sample Complexity
## Full Version

Haim Kaplan[*]      Yishay Mansour[†]      Uri Stemmer[‡]      Eliad Tsfadia[§]

October 22, 2020

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

## 1.3 Other Related Work

Dunagan and Vempala [2008] showed an efficient (non-private) learner for the linear feasibility problem that works in (a variant of) the *statistical query (SQ)* model of Kearns [1998]. It is known that algorithms

operating in the SQ model can be transformed to preserve differential privacy [Blum et al., 2005], and the algorithm of Dunagan and Vempala [2008] yields a differentially private efficient algorithm for solving the $(X, d, m)$-linear feasibility problem for $m \geq \text{poly}(d, \log |X|)$. Another related work is that of Hsu et al. [2014] who studied a variant of the linear feasibility problem with a certain large-margin assumption. Specifically, given a feasible collection of linear constraints of the form $\sum_{i=1}^d a_i x_i \geq 0$, their algorithm finds a solution $\mathbf{x}^*$ that approximately satisfies most of them (that is, $\sum_{i=1}^d a_i x_i^* \geq -c$ for most of the constraints, for a "margin parameter" $c > 0$). Large-margin assumptions were also utilized by Blum et al. [2005] and Nguyen et al. [2019] who designed efficient private learners for learning large-margin halfspaces. In addition, several other works developed tools that implicitly imply private learning of large-margin halfspaces, such as the works of Chaudhuri et al. [2011] and Bassily et al. [2014]. We remark that in this work we do not make large-margin assumptions.

# 2 Preliminaries

In this section we state basic preliminaries from learning theory and differential privacy, introduce a tool that enables our constructions, describe the geometric objects we use throughout the paper, and present some of their properties.

**Notations.** We use calligraphic letters to denote sets and boldface for vectors and matrices. We let $\mathbb{N}_0 = \mathbb{N} \cup \{0\}$. For $\mathbf{x} = (x_1, \dots, x_d) \in \mathbb{R}^d$ and $\mathbf{y} = (y_1, \dots, y_d) \in \mathbb{R}^d$, we let $\langle \mathbf{x}, \mathbf{y} \rangle := \sum_{i=1}^d x_i y_i$ be the inner-product of $\mathbf{x}$ and $\mathbf{y}$, and $\|\mathbf{x}\| := \sqrt{\langle \mathbf{x}, \mathbf{x} \rangle}$ be the norm of $\mathbf{x}$. For two integers $a \leq b$, let $[[a, b]] := \{a, a+1, \dots, b\}$ and let $[[\pm a]] := [[-|a|, |a|]]$. Given sets $\mathcal{S}_1, \dots, \mathcal{S}_k$ and $k$-input function $f$, let $f(\mathcal{S}_1, \dots, \mathcal{S}_k) := \{f(x_1, \dots, x_j) \colon x_i \in \mathcal{S}_i\}$, e.g., $[[\pm 5]]/[[7, 20]] = \{x/y \colon x \in [[\pm 5]], y \in [[7, 20]]\}$. Given a set $\mathcal{X}$ we let $\mathcal{X}^*$ be the set of all possible multisets whose elements are taken (possibly with repetitions) from the set $\mathcal{X}$.

## 2.1 Preliminaries from Differential Privacy

Consider a database where each record contains information of an individual. An algorithm is said to preserve differential privacy if a change of a single record of the database (i.e., information of an individual) does not significantly change the output distribution of the algorithm. Intuitively, this means that the information inferred about an individual from the output of a differentially-private algorithm is similar to the information that would be inferred had the individual's record been arbitrarily modified or removed. Formally:

**Definition 2.1** (Differential privacy [Dwork et al., 2006b,a]). *A randomized algorithm $\mathcal{A}$ is $(\varepsilon, \delta)$-differentially private if for all neighboring databases $S_1, S_2$ (i.e., differ by exactly one entry), and for all sets $\mathcal{F}$ of outputs,*

$$\Pr[\mathcal{A}(S_1) \in \mathcal{F}] \leq \exp(\varepsilon) \cdot \Pr[\mathcal{A}(S_2) \in \mathcal{F}] + \delta, \tag{1}$$

*where the probability is taken over the random coins of $\mathcal{A}$. When $\delta = 0$ we omit it and say that $\mathcal{A}$ preserves $\varepsilon$-differential privacy.*

We use the term *pure* differential privacy when $\delta = 0$ and the term *approximate* differential privacy when $\delta > 0$, in which case $\delta$ is typically a negligible function of the database size $m$.

We will later present algorithms that access their input database using (several) differentially private algorithms. We will use the following composition theorems.

**Theorem 2.2** (Basic composition). *If $\mathcal{A}_1$ and $\mathcal{A}_2$ satisfy $(\varepsilon_1, \delta_1)$ and $(\varepsilon_2, \delta_2)$ differential privacy, respectively, then their concatenation $\mathcal{A}(\mathcal{S}) = \langle \mathcal{A}_1(\mathcal{S}), \mathcal{A}_2(\mathcal{S}) \rangle$ satisfies $(\varepsilon_1 + \varepsilon_2, \delta_1 + \delta_2)$-differential privacy.*

Moreover, a similar theorem holds for the adaptive case, where an algorithm uses $k$ *adaptively chosen* differentially private algorithms (that is, when the choice of the next differentially private algorithm that is used depends on the outputs of the previous differentially private algorithms).

**Theorem 2.3** ([Dwork et al., 2006a, Dwork and Lei, 2009b]). *An algorithm that adaptively uses $k$ algorithms that preserves $(\varepsilon/k, \delta/k)$-differential privacy (and does not access the database otherwise) ensures $(\varepsilon, \delta)$-differential privacy.*

Note that the privacy guaranties of the above bound deteriorates linearly with the number of interactions. By bounding the *expected* privacy loss in each interaction (as opposed to worst-case), Dwork et al. [2010] showed the following stronger composition theorem, where privacy deteriorates (roughly) as $\sqrt{k}\varepsilon + k\varepsilon^2$ (rather than $k\varepsilon$).

**Theorem 2.4** (Advanced composition Dwork et al. [2010], restated). *Let $0 < \varepsilon_0, \delta' \leq 1$, and let $\delta_0 \in [0, 1]$. An algorithm that adaptively uses $k$ algorithms that preserves $(\varepsilon_0, \delta_0)$-differential privacy (and does not access the database otherwise) ensures $(\varepsilon, \delta)$-differential privacy, where $\varepsilon = \sqrt{2k \ln(1/\delta')} \cdot \varepsilon_0 + 2k\varepsilon_0^2$ and $\delta = k\delta_0 + \delta'$.*

## 2.2 Preliminaries from Learning Theory

We next define the probably approximately correct (PAC) model of Valiant [1984]. A concept $c : \mathcal{X} \to \{0, 1\}$ is a predicate that labels *examples* taken from the domain $\mathcal{X}$ by either 0 or 1. A *concept class* $\mathcal{C}$ over $\mathcal{X}$ is a set of concepts (predicates) mapping $\mathcal{X}$ to $\{0, 1\}$. A learning algorithm is given examples sampled according to an unknown probability distribution $\mu$ over $\mathcal{X}$, and labeled according to an unknown *target* concept $c \in \mathcal{C}$. The learning algorithm is successful when it outputs a hypothesis $h$ that approximates the target concept over samples from $\mu$. More formally:

**Definition 2.5.** The *generalization error* of a hypothesis $h : X \to \{0, 1\}$ is defined as

$$\text{error}_\mu(c, h) = \Pr_{x \sim \mu}[h(x) \neq c(x)].$$

If $\text{error}_\mu(c, h) \leq \alpha$ we say that $h$ is $\alpha$-*good* for $c$ and $\mu$.

**Definition 2.6** (PAC Learning [Valiant, 1984]). Algorithm $\mathcal{A}$ is an $(\alpha, \beta, m)$-*PAC learner* for a concept class $\mathcal{C}$ over $\mathcal{X}$ using hypothesis class $\mathcal{H}$ if for all concepts $c \in \mathcal{C}$, all distributions $\mu$ on $\mathcal{X}$, given an input of $m$ samples $S = (z_1, \dots, z_m)$, where $z_i = (x_i, c(x_i))$ and each $x_i$ is drawn i.i.d. from $\mu$, algorithm $\mathcal{A}$ outputs a hypothesis $h \in \mathcal{H}$ satisfying

$$\Pr[\text{error}_\mu(c, h) \leq \alpha] \geq 1 - \beta,$$

where the probability is taken over the random choice of the examples in $S$ according to $\mu$ and the random coins of the learner $\mathcal{A}$. If $\mathcal{H} \subseteq \mathcal{C}$ then $\mathcal{A}$ is called a *proper* PAC learner; otherwise, it is called an *improper* PAC learner.

**Definition 2.7.** For a labeled sample $S = (x_i, y_i)_{i=1}^m$, the *empirical error* of $h$ is

$$\text{error}_S(h) = \frac{1}{m}|\{i : h(x_i) \neq y_i\}|.$$

We use the following fact.

**Theorem 2.8** (Blumer et al. [1989]). *Let $\mathcal{C}$ and $\mu$ be a concept class and a distribution over a domain $\mathcal{X}$. Let $\alpha, \beta > 0$, and $m \geq \frac{48}{\alpha}\left(10VC(\mathcal{C})\log\left(\frac{48e}{\alpha}\right) + \log\left(\frac{5}{\beta}\right)\right)$. Suppose that we draw a sample $\mathcal{S} = (x_i)_{i=1}^m$, where each $x_i$ is drawn i.i.d. from $\mu$. Then*

$$\Pr\left[\exists c, h \in \mathcal{C} \text{ s.t. } \text{error}_\mu(c, h) \geq \alpha \text{ and } \text{error}_\mathcal{S}(c, h) \leq \alpha/10\right] \leq \beta.$$

## 2.3 Private Learning

Consider a learning algorithm $\mathcal{A}$ in the probably approximately correct (PAC) model of Valiant [1984]. We say that $\mathcal{A}$ is a *private* learner if it also satisfies differential privacy w.r.t. its training data. Formally,

**Definition 2.9** (Private PAC Learning [Kasiviswanathan et al., 2011]). Let $\mathcal{A}$ be an algorithm that gets an input $S = (z_1, \ldots, z_m)$. Algorithm $\mathcal{A}$ is an $(\varepsilon, \delta)$-*differentially private* $(\alpha, \beta)$-*PAC learner with sample complexity* $m$ for a concept class $\mathcal{C}$ over $\mathcal{X}$ using hypothesis class $\mathcal{H}$ if

PRIVACY. Algorithm $\mathcal{A}$ is $(\varepsilon, \delta)$-differentially private (as in Definition 1.1);

UTILITY. Algorithm $\mathcal{A}$ is an $(\alpha, \beta)$-*PAC learner* for $\mathcal{C}$ with sample complexity $m$ using hypothesis class $\mathcal{H}$.

Note that the utility requirement in the above definition is an average-case requirement, as the learner is only required to do well on typical samples (i.e., samples drawn i.i.d. from a distribution $\mu$ and correctly labeled by a target concept $c \in \mathcal{C}$). In contrast, the privacy requirement is a worst-case requirement, that must hold for every pair of neighboring databases (no matter how they were generated, even if they are not consistent with any concept in $\mathcal{C}$).

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

 \colon\ f(\mathcal{S}, \mathbf{x}) \geq \Delta \cdot f_{\text{Conv}}(\mathcal{S}, \mathbf{x}) - (\Delta - 1) \cdot M_{\mathcal{S}}$

Requirement 3.2 can be interpreted as follows: For any $(\mathcal{S}, \mathbf{x}) \in \mathcal{X}^* \times \mathbb{R}^d$, if $f_{\text{Conv}}(\mathcal{S}, \mathbf{x}) = (1 - \lambda)M_{\mathcal{S}}$, then $f(\mathcal{S}, \mathbf{x}) \geq (1 - \lambda\Delta)M_{\mathcal{

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

} = \frac{\alpha}{2d\Delta}, \tilde{\beta} = \frac{\beta}{d}, \tilde{\varepsilon} = \frac{\varepsilon}{2\sqrt{2d\ln(2/\delta)}}, \tilde{\delta} = \frac{\delta}{2d}$.
   Let $x_i^*$ be its output.

(iii) Return $\mathbf{x}^* = (x_1^*, \dots, x_d^*)$.

---

Figure 1: Algorithm for finding a point $\mathbf{x}^* \in \mathbb{R}^d$ with $f(\mathcal{S}, \mathbf{x}^*) \geq (1 - \alpha)M_{\mathcal{S}}$.

**Utility.** We prove by induction that after step $i$ of the algorithm, with probability at least $1 - i\beta/d$, the returned values $x_1^*, \dots, x_i^*$ satisfy $Q_{x_1^*, \dots, x_{i-1}^*}(\mathcal{S}, x_i^*) \geq (1 - \frac{\alpha}{2d\Delta})^i M_{\mathcal{S}}$, i.e., there are $x_{i+1}, \dots, x_d \in \mathbb{R}$ such that $f_{\text{Conv}}(\mathcal{S}, x_1^*, \dots, x_i^*, x_{i+1}, \dots, x_d) \geq (1 - \frac{\alpha}{2d\Delta})^i |\mathcal{S}|$. This concludes the utility part since after the $d$ iterations, with probability $1 - \beta$, $\mathcal{A}_{\text{OptimizeHighDimFunc}}$ outputs a point $\mathbf{x}^*$ with $f_{\text{Conv}}(\mathcal{S}, \mathbf{x}^*) \geq (1 - \frac{\alpha}{2d\Delta})^d M_{\mathcal{S}} \geq (1 - \frac{\alpha}{\Delta})M_{\mathcal{S}}$ (follows by the inequality $1 - x/2 \geq e^{-x}$ for $x \in [0,1]$), and by assumption on $\Delta$ we deduce that $f(\mathcal{S}, \mathbf{x}^*) \geq (1 - \alpha)M_{\mathcal{S}}$.

The basis of the induction is $i = 1$: By the assumption on $\left\{ \tilde{\mathcal{X}}_i(\cdot) \right\}_{i=1}^d$, there exists a value in $\tilde{\mathcal{X}}_1$ that maximize $Q(\mathcal{S}, \cdot)$. By Proposition 2.12 along with the assumption on $M_{\mathcal{S}}$, $\mathcal{A}_{\text{RecConcave}}$ finds with probability at least $1 - \beta/d$ a point $x_1^* \in \tilde{\mathcal{X}}_1$ with $Q(\mathcal{S}, x_1^*) \geq (1 - \frac{\alpha}{2d\Delta})M_{\mathcal{S}}$.

Next, by the induction hypothesis for $i - 1$, it holds that $\max_{x_i \in \mathbb{R}} \left\{ Q_{x_1^*, \dots, x_{i-1}^*}(\mathcal{S}, x_i) \right\} \geq (1 - \frac{\alpha}{2d\Delta})^{i-1} M_{\mathcal{S}}$ with probability at least $1 - (i-1)\beta/d$, and recall that by assumption there exists $x_i \in \tilde{\mathcal{X}}_i = \tilde{\mathcal{X}}_i(x_1^*, \dots, x_{i-1}^*)$ that reaches the maximum of $Q_{x_1^*, \dots, x_{i-1}^*}(\mathcal{S}, \cdot)$. Therefore, by Proposition 2.12 along with the assumption on $M_{\mathcal{S}}$, with probability at least $(1 - \beta/d)(1 - (i-1)\beta/d) \geq 1 - i\beta/d$, Algorithm $\mathcal{A}_{\text{RecConcave}}$ returns $x_i^* \in \tilde{\mathcal{X}}_i$ with $Q_{x_1^*, \dots, x_{i-1}^*}(\mathcal{S}, x_i^*) \geq (1 - \frac{\alpha}{2d\Delta})^i M_{\mathcal{S}}$.

**Privacy.** By Proposition 2.12 and Claim 3.5, each invocation of $\mathcal{A}_{\text{OptimizeHighDimFunc}}$ is $(\tilde{\varepsilon}, \tilde{\delta})$-differentially private. $\mathcal{A}_{\text{OptimizeHighDimFunc}}$ invokes $\mathcal{A}_{\text{RecConcave}}$ $d$ times. Thus, by Theorem 2.4 (the advanced composition) with $\delta' = \delta/2$, it follows that $\mathcal{A}_{\text{OptimizeHighDimFunc}}$ is $(\frac{\varepsilon}{2} + \frac{\varepsilon^2}{4\ln(2/\delta)}, \delta)$ differentially-private, which implies $(\varepsilon, \delta)$-privacy whenever $\varepsilon \leq 1$ and $\delta \leq 1/2$. $\qquad \square$

## 4 The Linear Feasibility Problem

In this section we show how the method from Section 3 can be used for privately approximating the linear feasibility problem. In this problem, we are given a finite grid $\mathcal{X} = [[\pm X]] := \{x \in \mathbb{Z} \colon |x| \leq X\}$ for some $X \in \mathbb{N}$ and a dataset $\mathcal{S} \in (\mathcal{X}^d \times \mathcal{X})^*$ such that each $(\mathbf{a}, w) \in \mathcal{S}$ represents the linear constraint $\langle \mathbf{a}, \mathbf{x} \rangle \geq w$ which defines the halfspace $\text{hs}_{\mathbf{a}, w}$ in $\mathbb{R}^d$. In the following, we let $\text{depth}_{\mathcal{S}}(\mathbf{x}) := |\{(\mathbf{a}, w) \in \mathcal{S} \colon \mathbf{x} \in \text{hs}_{\mathbf{a}, w}\}|$ (that is, the number of halfspaces in $\mathcal{S}$ that contain the point $\mathbf{x}$). Our goal is to describe, given $\alpha, \beta, \varepsilon, \delta \in (0,1)$, an

$(\varepsilon, \delta)$-differential private algorithm that satisfies the following utility guarantee: Given a realizable dataset of halfspaces (i.e., there exists a point $\mathbf{x} \in \mathbb{R}^d$ with $\text{depth}_{\mathcal{S}}(\mathbf{x}) = |\mathcal{S}|$), then with probability $1 - \beta$ the algorithm should output a point $\mathbf{x}^*$ with $\text{depth}_{\mathcal{S}}(\mathbf{x}^*) \geq (1 - \alpha) |\mathcal{S}|$.

In the following, let cdepth be the convexification of the function depth (according to Definition 3.1). That is, $\text{cdepth}_{\mathcal{S}}(\mathbf{x}) = f_{\text{Conv}}(\mathcal{S}, \mathbf{x})$ for the function $f(\mathcal{S}, \mathbf{x}) = \text{depth}_{\mathcal{S}}(\mathbf{x})$. As a first step towards applying Theorem 3.7 for maximizing depth, we need to determine a value $\Delta \geq 1$ that satisfies Requirement 3.2. Namely, we need to lower bound $\text{depth}_{\mathcal{S}}(\mathbf{x})$ in terms of $\text{cdepth}_{\mathcal{S}}(\mathbf{x})$ and $M_{\mathcal{S}} = |\mathcal{S}|$. For that, we prove the following claim.

**Claim 4.1.** *For any $\mathcal{S} \in (\mathbb{R}^d \times \mathbb{R})^*$ and any $\mathbf{x} \in \mathbb{R}^d$, it holds that*

$$\text{depth}_{\mathcal{S}}(\mathbf{x}) \geq (d + 1) \cdot \text{cdepth}_{\mathcal{S}}(\mathbf{x}) - d |\mathcal{S}|.$$

*Proof.* Fix $\mathcal{S} \in (\mathbb{R}^d \times \mathbb{R})^*$ and $\mathbf{x} \in \mathbb{R}^d$, and let $k = \text{cdepth}(\mathbf{x})$. By definition it holds that $\mathbf{x} \in \text{ConvexHull}(\mathcal{D}_{\mathcal{S}}(k))$ for $\mathcal{D}_{\mathcal{S}}(k) = \{\mathbf{x}' \colon \text{depth}_{\mathcal{S}}(\mathbf{x}') \geq k\}$. Therefore, by Caratheodory's theorem (Fact 2.15) it holds that $\mathbf{x}$ is a convex combination of at most $d + 1$ points $\mathbf{x}_1, \ldots, \mathbf{x}_{d+1} \in \mathcal{D}_{\mathcal{S}}(k)$. In the following, for $\mathbf{x}' \in \mathbb{R}^d$ let $\mathcal{T}_{\mathbf{x}'} := \{(\mathbf{a}, w) \in \mathcal{S} \colon \mathbf{x}' \notin \text{hs}_{\mathbf{a}, w}\}$ and observe that $\text{depth}_{\mathcal{S}}(\mathbf{x}') = |\mathcal{S}| - |\mathcal{T}_{\mathbf{x}'}|$. Therefore, because for all $i \in [d+1]$ we have $\text{depth}(\mathbf{x}_i) \geq k$, it holds that $|\mathcal{T}_{\mathbf{x}_i}| \leq |\mathcal{S}| - k$. Furthermore, note that $\mathcal{T}_{\mathbf{x}} \subseteq \bigcup_{i=1}^{d} \mathcal{T}_{\mathbf{x}_i}$ (holds since each halfspace that contains a set of points also contains any convex combination of them). We conclude that $\text{depth}_{\mathcal{S}}(\mathbf{x}) \geq |\mathcal{S}| - \sum_{i=1}^{d+1} |\mathcal{T}_{\mathbf{x}_i}| \geq |\mathcal{S}| - (d+1)(|\mathcal{S}| - k) = (d+1)k - d |\mathcal{S}|$. $\square$

Namely, $\Delta = d + 1$ satisfies Requirement 3.2 for the function $f(\mathcal{S}, \mathbf{x}) = \text{depth}_{\mathcal{S}}(\mathbf{x})$.
The second step towards applying Theorem 3.7 is to determine an iterative sequence of finite domains $\left\{ \tilde{\mathcal{X}}_i(\cdot) \right\}_{i=1}^{d}$ that satisfies Requirement 3.6. Namely, our goal is to determine a finite domain $\tilde{\mathcal{X}}_i = \tilde{\mathcal{X}}_i(x_1^*, \ldots, x_{i-1}^*)$ such that there exists $x_i^* \in \tilde{\mathcal{X}}_i$ that reaches the maximum of $Q_{x_1^*, \ldots, x_{i-1}^*}(\mathcal{S}, \cdot)$ under $\mathbb{R}$, where $Q_{x_1^*, \ldots, x_{i-1}^*}$ is defined below.

**Definition 4.2.** For every $1 \leq i \leq d$ and every $x_1^*, \ldots, x_{i-1}^* \in \mathbb{R}$, define

$$Q_{x_1^*, \ldots, x_{i-1}^*}(\mathcal{S}, x_i) := \max_{\tilde{x}_{i+1}, \ldots, \tilde{x}_d \in \mathbb{R}} \text{cdepth}_{\mathcal{S}}(x_1^*, \ldots, x_{i-1}^*, x_i, \tilde{x}_{i+1}, \ldots, \tilde{x}_d).$$

The following lemma, proven in Appendix B.1, states that at least one of the maximum points $x_i^*$ can be derived by solving a system of linear equations with bounded coefficients.

**Lemma 4.3.** *Let $X \in \mathbb{N}$, $\mathcal{X} = [[\pm X]]$, $\mathcal{S} \in (\mathcal{X}^d \times \mathcal{X})^*$, $i \in [d]$, let $x_1^*, \ldots, x_{i-1}^* \in \mathbb{R}$ and let $Q_{x_1^*, \ldots, x_{i-1}^*}$ be the function from Definition 4.2. Then there exists an invertible matrix $\mathbf{A} \in \mathcal{X}^{(d-i+1) \times (d-i+1)}$ and values*

$$b_i, \ldots, b_d \in \mathcal{X} - \sum_{j=1}^{i-1} x_j^* \cdot \mathcal{X} := \bigcup_{w, a_1, \ldots, a_{i-1} \in \mathcal{X}} \left\{ w - \sum_{j=1}^{i-1} a_j x_j^* \right\}$$

*such that $(x_i^*, \ldots, x_d^*)^T := \mathbf{A}^{-1} \cdot (b_i, \ldots, b_d)^T$ satisfies*

$$\text{cdepth}_{\mathcal{S}}(x_1^*, \ldots, x_d^*) = Q_{x_1^*, \ldots, x_{i-1}^*}(\mathcal{S}, x_i^*) = \max_{x_i \in \mathbb{R}} \left\{ Q_{x_1^*, \ldots, x_{i-1}^*}(\mathcal{S}, x_i) \right\}.$$

Using Lemma 4.3, we can now define a finite domain for each iteration $i \in [d]$.

**Definition 4.4** (The domain $\tilde{\mathcal{X}}_i = \tilde{\mathcal{X}}_i(x_1^*, \ldots, x_{i-1}^*)$)**.** We define the domains $\left\{ \tilde{\mathcal{X}}_i \right\}_{i=1}^{d}$ iteratively. For $i = 1$ let $\tilde{\mathcal{X}}_1 := \tilde{\mathcal{X}}_1' / \tilde{\mathcal{X}}_1''$ where $\tilde{\mathcal{X}}_1' := [[\pm(d \cdot d!) \cdot X^d]]$ and $\tilde{\mathcal{X}}_1'' := ([[\pm d! \cdot X^d]]) \setminus \{0\}$.[7] For $i > 1$ and given $x_j^* = s_j / t_j \in \tilde{\mathcal{X}}_j' / \tilde{\mathcal{X}}_j'' = \tilde{\mathcal{X}}_j$ for $j \in [i-1]$, define $\tilde{\mathcal{X}}_i = \tilde{\mathcal{X}}_i(x_1^*, \ldots, x_{i-1}^*) := \tilde{\mathcal{X}}_i' / \tilde{\mathcal{X}}_i''$ where $\tilde{\mathcal{X}}_i' := [[\pm(d \cdot d!)^i \cdot X^{di}]]$ and $\tilde{\mathcal{X}}_i'' = \tilde{\mathcal{X}}_i''(t_{i-1}) := ([[\pm d! \cdot X^d]] \cdot t_{i-1}) \setminus \{0\}$.

We next prove that the above sequence $\left\{\tilde{\mathcal{X}}_i(\cdot)\right\}_{i=1}^d$ satisfies Requirement 3.6.

**Lemma 4.5.** *Let $X \in \mathbb{N}$, $\mathcal{X} = [[\pm X]]$, $\mathcal{S} \in (\mathcal{X}^d \times \mathcal{X})^*$, $i \in [d]$ and $x_1^* \in \tilde{\mathcal{X}}_1, \ldots, x_{i-1}^* \in \tilde{\mathcal{X}}_{i-1}$, where $\tilde{\mathcal{X}}_j = \tilde{\mathcal{X}}_j(x_1^*, \ldots, x_{i-1}^*)$, for $j \in [i]$, is according to Definition 4.4. Then there exists $x_i^* \in \tilde{\mathcal{X}}_i$ such that*

$$Q_{x_1^*, \ldots, x_{i-1}^*}(\mathcal{S}, x_i^*) = \max_{x_i \in \mathbb{R}} \left\{ Q_{x_1^*, \ldots, x_{i-1}^*}(\mathcal{S}, x_i) \right\} \tag{3}$$

*Proof.* We are given $x_1^* \in \tilde{\mathcal{X}}_1, \ldots, x_{i-1}^* \in \tilde{\mathcal{X}}_{i-1}$ such for all $j \in [i-1]$: $x_j^* = s_j/t_j$ for some $s_j \in [[\pm(d \cdot d!)^j \cdot X^{dj}]]$ and $t_j \in \left( [[\pm d! \cdot X^d]] \cdot t_{j-1} \right) \setminus \{0\}$ (letting $t_0 = 1$), and our goal is to prove the existence of $x_i^* \in \tilde{\mathcal{X}}_i$ that satisfies Equation (3). By Lemma 4.3, there exist an invertible matrix $\mathbf{A} \in \mathcal{X}^{(d-i+1) \times (d-i+1)}$ and values $b_i, \ldots, b_d$ with

$$\begin{aligned}
b_j \in \mathcal{X} - \sum_{j=1}^{i-1} x_j^* \cdot \mathcal{X} &= \frac{t_{i-1} \cdot \mathcal{X} - \sum_{j=1}^{i-1} s_j \cdot (t_{i-1}/t_j) \cdot \mathcal{X}}{t_{i-1}} \\
&\in \frac{[[\pm d!^{i-1} \cdot X^{d(i-1)+1}]] + \sum_{j=1}^{i-1} [[\pm(d \cdot d!)^j \cdot X^{dj}]] \cdot [[\pm d!^{i-1-j} \cdot X^{d(i-1-j)}]] \cdot [[\pm X]]}{t_{i-1}} \\
&\in \frac{1}{t_{i-1}} \cdot [[\pm d^i \cdot d!^{i-1} \cdot X^{d(i-1)+1}]],
\end{aligned}$$

such that the unique solution $(x_i^*, \ldots, x_d^*)$ to the system of linear equations $\mathbf{A}(x_i, \ldots, x_d)^T = (b_i, \ldots, b_d)^T$ satisfies $Q_{x_1^*, \ldots, x_{i-1}^*}(\mathcal{S}, x_i^*) = \max_{x_i \in \mathbb{R}} \left\{ Q_{x_1^*, \ldots, x_{i-1}^*}(\mathcal{S}, x_i) \right\}$. Hence, we deduce by Cramer's rule that

$$\begin{aligned}
x_i^* = \frac{\det(\tilde{\mathbf{A}})}{\det(\mathbf{A})} &\in \frac{\sum_{j=i}^d b_j \cdot [[\pm(d-1)! \cdot X^{d-i}]]}{[[\pm d! \cdot X^{d-i+1}]] \setminus \{0\}} \\
&\subseteq \frac{1}{t_{i-1}} \cdot [[\pm d^i \cdot d!^{i-1} \cdot X^{d(i-1)+1}]] \cdot \frac{[[\pm d! \cdot X^{d-i}]]}{[[\pm d! \cdot X^{d-i+1}]] \setminus \{0\}} \subseteq \tilde{\mathcal{X}}_i,
\end{aligned}$$

where $\tilde{\mathbf{A}}$ is the matrix $\mathbf{A}$ when replacing its first column with $(b_i, \ldots, b_d)^T$. $\square$

## 4.1 The Algorithm

In Figure 2, we present an $(\varepsilon, \delta)$-differentially private algorithm $\mathcal{A}_{\text{FindDeepPoint}}$ that given a realizable dataset of halfspaces $\mathcal{S}$, finds with probability at least $1 - \beta$ a point whose depth is at least $(1 - \alpha) |\mathcal{S}|$.

---

**Algorithm $\mathcal{A}_{\text{FindDeepPoint}}$**

(i) Let $\alpha, \beta, \varepsilon, \delta \in (0,1)$ be the utility/privacy parameters, and let $\mathcal{S} \in \left( \mathcal{X}^d \times \mathcal{X} \right)^*$ be an input dataset.

(ii) Execute $\mathcal{A}_{\text{OptimizeHighDimFunc}}$ on the function $f(\mathcal{S}, \cdot) = \text{depth}_{\mathcal{S}}(\cdot)$, with parameters $\alpha, \beta, \varepsilon, \delta$, $\Delta = d + 1$ and the sequence $\left\{ \tilde{\mathcal{X}}_i(\cdot) \right\}_{i=1}^d$ defined in Definition 4.4.

(ii) Output the resulting point $\mathbf{x}^*$.

---

Figure 2: Algorithm $\mathcal{A}_{\text{FindDeepPoint}}$ for finding a point $\mathbf{x}^* \in \mathbb{R}^d$ with $\text{depth}_{\mathcal{S}}(\mathbf{x}^*) \geq (1 - \alpha) |\mathcal{S}|$.

**Theorem 4.6** (Restatement of Theorem 1.4). *Let $\alpha, \beta, \varepsilon \leq 1$, $\delta < 1/2$, $X \in \mathbb{N}$, $\mathcal{X} = [[\pm X]]$ and let $\mathcal{S} \in \left( \mathcal{X}^d \times \mathcal{X} \right)^*$ be a realizable dataset of halfspaces with*

$$|\mathcal{S}| = O\left( d^{2.5} \cdot 2^{O(\log^* X + \log^* d)} \frac{\log^{1.5}\left(\frac{1}{\delta}\right) \log\left(\frac{d}{\beta}\right)}{\varepsilon \alpha} \right).$$

*Then, $\mathcal{A}_{\mathrm{FindDeepPoint}}$ is an $(\varepsilon, \delta)$-differentially private algorithm that with probability at least $1 - \beta$ returns a point $\mathbf{x}^* \in \mathbb{R}^d$ with $\mathrm{depth}(\mathbf{x}^*) \geq (1 - \alpha) |\mathcal{S}|$. Furthermore, $\mathcal{A}_{\mathrm{FindDeepPoint}}$ runs in time*

$$T = \mathrm{poly}(d) \cdot |\mathcal{S}| \cdot \left( |\mathcal{S}|^d \cdot \log X + \mathrm{polylog}(1/\alpha, 1/\beta, 1/\varepsilon, 1/\delta, X) \right).$$

Since depth is a sensitivity-1 function, the proof of Theorem 4.6 immediately follow by Theorem 3.7, Claim 4.1 and Lemma 4.5. See Appendix B.2 for the running time analysis.

# 5  Learning Halfspaces

In this section we describe our private empirical risk minimization (ERM) learner of halfspaces, and at the end we state our (almost) immediate corollary about private PAC learning.

In the considered problem, we are given a finite grid $\mathcal{X} = [[\pm X]] := \{x \in \mathbb{Z} : |x| \leq X$ for some $X \in \mathbb{N}$ and a dataset of labeled points $\mathcal{S} \in (\mathcal{X}^d \times \{-1, 1\})^*$. We say that $\mathcal{S}$ is a realizable dataset of points if there exists $(\mathbf{a}, w) \in \mathbb{R}^d \times \mathbb{R}$ with $\mathrm{error}_{\mathcal{S}}(c_{\mathbf{a},w}) := |\{(\mathbf{x}, y) \in \mathcal{S} : c_{\mathbf{a},w}(\mathbf{x}) \neq y\}| / |\mathcal{S}| = 0$, letting $c_{\mathbf{a},w} \colon \mathcal{X}^d \mapsto \{-1, 1\}$ be the concept function that outputs 1 iff $\mathbf{x} \in \mathrm{hs}_{\mathbf{a},w}$. Our goal is to describe, given $\alpha, \beta, \varepsilon, \delta \in (0, 1)$, an $(\varepsilon, \delta)$-differential private algorithm that satisfies the following utility guarantee: Given a realizable dataset of points $\mathcal{S}$, the algorithm should output with probability $1 - \beta$ a pair $(\mathbf{a}^*, w^*)$ with $\mathrm{error}_{\mathcal{S}}(c_{\mathbf{a}^*, w^*}) \leq \alpha$

## 5.1  A Reduction to the Linear Feasibility Problem

We reduce the problem of learning a halfspace to the linear feasibility problem by using geometric duality between points and halfspaces. Formally, we translate a halfspace $\mathrm{hs}_{\mathbf{a},w}$ to the point $(\mathbf{a}, w) \in \mathbb{R}^{d+1}$, and translate a labeled point $(\mathbf{x}, y) \in \mathcal{S}$ to the $(d+1)$-dimensional halfspace $\mathrm{hs}_{(y \cdot \mathbf{x}, -y), 0}$ which equals to $\{(\mathbf{a}, w) \in \mathbb{R}^{d+1} : \langle \mathbf{a}, \mathbf{x} \rangle \geq w\}$ if $y = 1$, and to $\{(\mathbf{a}, w) \in \mathbb{R}^{d+1} : \langle \mathbf{a}, \mathbf{x} \rangle \leq w\}$ if $y = -1$. By definition, for any realizable dataset of points $\mathcal{S}$, the multiset $\mathcal{S}' = \{((y \cdot \mathbf{x}, -y), 0) : (\mathbf{x}, y) \in \mathcal{S}\}$ is a realizable dataset of halfspaces. Therefore, by applying $\mathcal{A}_{\mathrm{FindDeepPoint}}$ on $\mathcal{S}'$ we obtain a deep point $(\mathbf{a}^*, w^*) \in \mathbb{R}^{d+1}$ for $\mathcal{S}'$, meaning that $\langle \mathbf{a}^*, y \cdot \mathbf{x} \rangle \geq y \cdot w^*$ for most of the $(\mathbf{x}, y) \in \mathcal{S}$, which is (almost) what we need. The problem is that the pairs $(\mathbf{x}, -1) \in \mathcal{S}$ with $\langle \mathbf{a}^*, \mathbf{x} \rangle = w^*$ do not count as points in $\mathrm{hs}_{\mathbf{a}^*, w^*}$ while they do count for the depth of $(\mathbf{a}^*, w^*)$ in $\mathcal{S}'$. Yet, assuming the points in $\mathcal{S}$ are in general position (an assumption that can be eliminated), then there can be at most $d$ such points.

## 5.2  The Algorithm

In Figure 3, we present our algorithm $\mathcal{A}_{\mathrm{LearnHalfSpace}}$ for learning halfspaces. Following the above intuition, the algorithm assumes that the points in $\mathcal{S}$ are in general position.

The following theorem summarizes the properties of $\mathcal{A}_{\mathrm{LearnHalfSpace}}$.

**Theorem 5.1** (Private ERM learner). *Let $\alpha, \beta, \varepsilon \leq 1$, $\delta < 1/2$, $X \in \mathbb{N}$, $\mathcal{X} = [[\pm X]]$ and let $\mathcal{S} \in \left( \mathcal{X}^d \times \{-1, 1\} \right)^*$ be a realizable dataset of points with $|\mathcal{S}| = O\left( d^{2.5} \cdot 2^{O(\log^* X + \log^* d)} \frac{\log^{1.5}\left(\frac{1}{\delta}\right) \log\left(\frac{d}{\beta}\right)}{\varepsilon \alpha} \right)$. $\mathcal{A}_{\mathrm{LearnHalfSpace}}$ is $(\varepsilon, \delta)$-differentially private. Moreover, assuming that the points in $\mathcal{S}$ are in general position,[8] then with probability $1 - \beta$ the algorithm returns a pair $(\mathbf{a}^*, w^*) \in \mathbb{R}^d \times \mathbb{R}$ with $\mathrm{error}_{\mathcal{S}}(c_{\mathbf{a}^*, w^*}) \leq \alpha$.*

*Proof.*

**Algorithm** $\mathcal{A}_{\text{LearnHalfSpace}}$

(i) Let $\alpha, \beta, \varepsilon, \delta \in (0,1)$ be the utility/privacy parameters, and let $\mathcal{S} \in \left(\mathcal{X}^d \times \{-1, 1\}\right)^*$ be an input dataset.

(ii) Execute $\mathcal{A}_{\text{FindDeepPoint}}$ on the multiset $\mathcal{S}' := \{((y \cdot \mathbf{x}, -y), 0) : (\mathbf{x}, y) \in \mathcal{S}\}$ and parameters $\alpha/2, \beta, \varepsilon, \delta$.

(ii) Output the resulting point $(\mathbf{a}^*, w^*) \in \mathbb{R}^{d+1}$.

---

Figure 3: Algorithm $\mathcal{A}_{\text{LearnHalfSpace}}$ for learning halfspaces.

**Utility.** Since $\mathcal{S}$ is a realizable dataset of points, it holds that $\mathcal{S}'$ is a realizable dataset of halfspaces. Therefore, by Theorem 4.6, it holds that with probability $1 - \beta$, algorithm $\mathcal{A}_{\text{FindDeepPoint}}$ finds $(\mathbf{a}^*, w^*) \in \mathbb{R}^{d+1}$ with $\text{depth}_{\mathcal{S}'}(\mathbf{a}^*, w^*) \geq (1 - \alpha/2)|\mathcal{S}|$, meaning that $\langle y \cdot \mathbf{x}, \mathbf{a}^* \rangle - y \cdot w^* \geq 0$ for $(1 - \alpha/2)|\mathcal{S}|$ of the pairs $(\mathbf{x}, y) \in \mathcal{S}$. Since the points in $\mathcal{S}$ are in general position, by the assumption on $|\mathcal{S}|$ there are at most $d < \alpha|\mathcal{S}|/2$ pairs $(\mathbf{x}, -1) \in \mathcal{S}$ that satisfy $\langle \mathbf{a}^*, \mathbf{x} \rangle = w^*$. Overall we obtain that $\text{error}_{\mathcal{S}}(c_{\mathbf{a}^*, w^*}) \leq (|\mathcal{S}| - \text{depth}_{\mathcal{S}'}(\mathbf{a}^*, w^*) + d)/|\mathcal{S}| \leq \alpha$.

**Privacy.** Follows by the privacy guarantee of $\mathcal{A}_{\text{FindDeepPoint}}$.

$\square$

We show how to remove the assumption that the points in $\mathcal{S}$ are in general position. Hence, since the VC dimension of $\texttt{HALFSPACE}(\mathbb{R}^d)$ is only $d + 1$, we immediately obtain a private PAC learner from our private ERM learner, which is the main result of this paper.

**Theorem 5.2** (Private PAC learner, restatement of Theorem 1.2)**.** *Let $\alpha, \beta, \varepsilon \leq 1$, $\delta < 1/2$, $X \in \mathbb{N}$ and let $\mathcal{X} = [[\pm X]]$. Then there exists an $(\varepsilon, \delta)$-differentially private $(\alpha, \beta)$-PAC learner with sample complexity $s$ for the class $\texttt{HALFSPACE}(\mathcal{X}^d)$ for $s = O\left(d^{2.5} \cdot 2^{O\left(\log^* X + \log^* d + \log^*\left(\frac{1}{\alpha\beta\varepsilon\delta}\right)\right)} \cdot \frac{\log^{1.5}\left(\frac{1}{\delta}\right) \log\left(\frac{d}{\alpha\beta}\right)}{\varepsilon\alpha}\right)$.*

The proof details of Theorem 5.2 appear at Appendix B.3. In the following section, we sketch the main technical challenges in the proof.

## 5.3 Proof Overview of Theorem 5.2

It is well known that given large enough dataset $\mathcal{S}$ of samples drawn i.i.d. from a distribution $\mu$ and labeled according to some concept function $c$, then for any hypothesis $h$, the empirical error of $h$ on $\mathcal{S}$ is close to the generalization error of $h$ on the distribution $\mu$ (see for example Theorem 2.8). Therefore, if $\mu$ is a distribution such that $s$ independent points from it are in general position with high probability, then by Theorem 5.1 we deduce that there exists a PAC leaning algorithm with small generalization error on $\mu$. However, the above argument does not hold for arbitrary distributions since Theorem 5.1 promises small empirical error only when the points in the dataset are in general position. In order to overcome this difficulty, given an $s$-size dataset $\mathcal{S}$, we first add a small random noise to each of the points in $\mathcal{S}$. To determine how much noise to add, we first prove in Lemma B.5 that the fact that the points are coming from a finite grid $\mathcal{X}^d = [[\pm X]]^d$ implies that there is a margin of at least $1/(d \cdot X)^{\text{poly}(d)}$ between the data points to a halfspace that agrees on all their labels. Moreover, in Lemma B.6 we determine the resolution of the noise that we need to take in order to guarantee general position with high probability. Now given an $s$-size dataset $\mathcal{S}$ drawn from $\mu$, we just add noise (independently) to each of the points in $\mathcal{S}$, where the size of the noise is smaller than the margin (to ensure that the noisy dataset remains realizable) and the resolution of the noise is high enough (to guarantee general position). This induces a (noisy) distribution $\tilde{\mu}$ that promises general position, and

now we are given a realizable dataset according to it. Therefore, we obtain a PAC learning algorithm with small generalization error on $\tilde{\mu}$. We end the proof by showing that every hypothesis that is good for $\tilde{\mu}$ is also good for $\mu$.

# 6 Open Questions

It is still remains open what is the minimal sample complexity that is required for learning halfspaces with an (approximate) differential privacy. Our work provides a new upper bound of $\approx d^{2.5} \cdot 2^{O(\log^* X)}$ which improves the state-of-the-art result of Beimel et al. [2019] by a $d^2$ factor, and improves the generic upper bound of Kasiviswanathan et al. [2011] whenever (roughly) $d < \log^2 X$.[9] Yet, there is still a gap from the best known lower bound of $\Omega(d \cdot \log^* X)$ for proper learning (Bun et al. [2015]) and $\Omega(d + \log^* X)$ for improper learning. In particular, it is still remains open whether we can avoid the exponential dependency in $\log^* X$ for $d > 1$. One option for answering it is by finding a different 1-dimensional quasi-concave optimization that only requires polynomial dependency in $\log^* X$, since RecConcave, the optimization that we are using, requires exponential dependency. Indeed, a recent work of Kaplan et al. [2020] shows an (almost) linear dependency in $\log^* X$ for 1-dimensional thresholds, which is a special case of a quasi-concave optimization, and it still remains open whether this result can be extended to the quasi-concave optimization case.

## Footnotes

[1]We remark that there is a loose characterization for private learning in terms of the *Littlestone dimension* Alon et al. [2019], Bun et al. [2020]. Specifically, these results state that the sample complexity of privately learning a class $C$ is somewhere between $\Omega(\log^* L)$ and $2^{O(L)}$, where $L$ is the Littlestone dimension of $C$. In our cont, for learning halfspaces, these results do not provide meaningful bounds on the sample complexity.

[2]A function $Q$ is *quasi-concave* if for any $x' \leq x \leq x''$ it holds that $Q(x) \geq \min\{Q(x'), Q(x'')\}$.

[3]The *sensitivity* of the function $f$ is the maximal difference by which the value of $f(\mathcal{S}, \mathbf{x})$ can change when modifying one element of the database $\mathcal{S}$. See Section 2 for a formal definition.

[4]For instance, consider the 2-dimensional constraints $x_2 \geq x_1$ and $x_2 \leq -x_1$. Then under the fixing $x_1^* = 1$, the depth of $x_2 = 0$ is 0 while the depth of $x_2 \in \{-1, 1\}$ is 1, yielding that $Q_{x_1^*}(0) < \min\left\{Q_{x_1^*}(-1), Q_{x_1^*}(1)\right\}$, and so $Q_{x_1^*}$ is not quasi-concave.

[5]We remark that the presentation here is oversimplified, and hides many of the challenges that arise in the actual analysis.

[6]We remark that this step might be involved for some $d$-dimensional functions, but is inherent for privately optimizing them (at least if the optimization is done coordinate by coordinate). Yet, once we determine such domains with some finite bound $\tilde{X}$ on their sizes, it usually not blows up the resulting sample complexity of our algorithm since it only depends on $2^{O(\log^* \tilde{X})}$ (see Theorem 3.7).

[7]Recall that for $a \in \mathbb{Z}^+$ we let $[[\pm a]] = \{-a, -a+1, \ldots, a\}$.

[8] A set of points in $\mathbb{R}^d$ are in general position if there are no $d + 1$ points that lie on the same hyperplane.

[9]We remark that even when $d > \log^2 X$, we offer significant improvements over the generic learner in terms of runtime. In particular, our algorithm runs in time (roughly) $n^d$, where $n$ is the number of samples, while the generic learner has a runtime of at least $X^{d^2}$.

[10]We remark that our description of this step is slightly oversimplified. Actually, in this step the algorithm partitions $\tilde{\mathcal{X}}$ into intervals $\{A_i\}$ and also into intervals $\{B_i\}$ that are right-shifted by $4 \cdot 2^k$. Then it is promised that in one of the partitions there is an interval that contains $P$.

[11]The projection of $\mathrm{hp}_{\mathbf{a}, w}$ to $V$ is simply define by setting $(x_1, \ldots, x_{i-1}) = (x_1^*, \ldots, x_{i-1}^*)$ to the equation $\sum_{j=1}^d a_j x_j = w$, which yields the $(d - i + 1)$-dimensional hyperplane $\sum_{j=i}^d a_j x_j = w - \sum_{j=1}^{i-1} a_j x_j^*$.

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

# A   Additional Details about Algorithm $\mathcal{A}_{\mathrm{RecConcave}}$

In this section we give a more detailed explanation on how $\mathcal{A}_{\mathrm{RecConcave}}$ works, but we refer to Beimel et al. [2013b] for the full details.

Fix a sensitivity-1 function $Q : \mathcal{X}^* \times \tilde{\mathcal{X}} \to \mathbb{R}$ and a database $\mathcal{S} \in \mathcal{X}^*$ such that $Q(\mathcal{S}, \cdot)$ is quasi-concave, and assume for simplicity that $\tilde{\mathcal{X}} = [X]$. Given the promise that there exists $m \in \tilde{\mathcal{X}}$ with $Q(\mathcal{S}, m) \geq r$, the task of $\mathcal{A}_{\mathrm{RecConcave}}$ is to find $\ell \in \tilde{\mathcal{X}}$ with $Q(\mathcal{S}, \ell) \geq (1 - \alpha)r$. Beimel et al. [2013b] defined the function

$$Q'(\mathcal{S}, j) := \min \left\{ L(\mathcal{S}, j) - (1 - \alpha)r, \ r - L(\mathcal{S}, j + 1) \right\}$$

for

$$L(\mathcal{S}, j) := \max_{a, b \in \tilde{\mathcal{X}}, b - a + 1 = 2^j} \left\{ \min_{i \in \{a, a+1, \ldots, b\}} Q(\mathcal{S}, i) \right\}.$$

Then they showed that $Q'(\mathcal{S}, \cdot)$ is quasi-concave and that there exists $j$ with $Q'(\mathcal{S}, j) \geq r'$ for $r' = \frac{\alpha}{2} r$. Therefore, by calling $\mathcal{A}_{\mathrm{RecConcave}}$ recursively (now on logarithmic size domain), we obtain a number $k$ with $Q'(\mathcal{S}, k) \geq (1 - \alpha)r'$. Now let $P_1$ be the $2 \cdot 2^k$ numbers before the maximum $m$, and $P_2$ be the $2 \cdot 2^k$ numbers after $m$. Since $L(\mathcal{S}, k+1) \leq r - q(\mathcal{S}, k) \leq \left(1 - \frac{\alpha}{4}\right) r$, it holds that each of $P_1$ and $P_2$ contains a point with $Q(\mathcal{S}, \cdot) \leq \left(1 - \frac{\alpha}{4}\right) r$. Therefore, since $Q(\mathcal{S}, \cdot)$ is quasi-concave, all the numbers outside $P = P_1 \cup P_2$ have $Q(\mathcal{S}, \cdot) \leq \left(1 - \frac{\alpha}{4}\right) r$. The algorithm now partitions $\tilde{\mathcal{X}}$ into intervals of size $8 \cdot 2^k$ such that one of them must contain $P$.[10] Then it chooses an interval using the algorithm of Thakurta and Smith [2013], which is an instantiation of the Propose-Test-Release framework (Dwork and Lei [2009a]), where the quality of an interval is the maximum attainable value of $Q(\mathcal{S}, \cdot)$ on it. Assuming that $r$ is large enough, the mechanism will choose the interval that contains $P$ with high probability. Since $L(\mathcal{S}, k) \geq q(\mathcal{S}, k) + (1 - \alpha)r \geq \left(1 - \frac{3\alpha}{4}\right) r$, there are $2^k$ points around $m$ that all have $Q(\mathcal{S}, \cdot) \geq \left(1 - \frac{3\alpha}{4}\right) r$. Hence, in the last step the algorithm defines 16 "equally spread" concepts inside the chosen $8 \cdot 2^k$-size segment and chooses one of them using the Exponential Mechanism (McSherry and Talwar [2007]).

## A.1   Running Time

We use the following fact about the running time of $\mathcal{A}_{\mathrm{RecConcave}}$.

**Fact A.1** (implicit in Beimel et al. [2016] (Remark 3.17))**.** The running time of $\mathcal{A}_{\mathrm{RecConcave}}$ on the function $Q : \mathcal{X}^* \times \tilde{\mathcal{X}} \to \mathbb{R}$ for $\tilde{\mathcal{X}} = [[\pm \tilde{X}]]$ and input parameters $\mathcal{S}, \alpha, \beta, \varepsilon, \delta$ is bounded by

$$(T_Q + T_L) \cdot \mathrm{polylog}(\tilde{X}, 1/\alpha, 1/\beta, 1/\varepsilon, 1/\delta),$$

where $T_Q$ is the time that takes to compute $Q(\mathcal{S}, i)$ (for every $i \in \tilde{\mathcal{X}}$), and $T_L$ is the time that takes to compute $L(\mathcal{S}, j) := \max_{a, b \in \tilde{\mathcal{X}}, b - a + 1 = 2^j} \left\{ \min_{i \in [[a, b]]} Q(\mathcal{S}, i) \right\}$.

# B   Missing Proofs

## B.1   Proving Lemma 4.3

In this section we prove Lemma 4.3. We start by defining a decreasing point for a function $Q$.

**Definition B.1** (decreasing point)**.** Let $Q \colon \mathbb{R} \mapsto \mathbb{R}$ be a function. We say that $x^* \in \mathbb{R}$ is a decreasing point for $Q$ if for all $x < x^*$ it holds that $Q(x) < Q(x^*)$, or for all $x > x^*$ it holds that $Q(x) < Q(x^*)$.

Note that for any dataset $\mathcal{S} \in (\mathcal{X}^d \times \mathcal{X})^*$ and any fixing of $x_1^*, \ldots, x_{i-1}^* \in \mathbb{R}$, the function $Q_{x_1^*, \ldots, x_{i-1}^*}(\mathcal{S}, \cdot)$ must have a decreasing point $x_i^*$ that reaches its maximum under $\mathbb{R}$ (unless the function is constant). The following lemma states that each decreasing point $x_i^*$ can be determined by intersection of hyperplanes in $\mathcal{S}$ under the subspace $\{\mathbf{x} \in \mathbb{R}^d \colon (x_1, \ldots, x_{i-1}) = (x_1^*, \ldots, x_{i-1}^*)\}$. Furthermore, there exists such intersection in which all the points that belongs to it have the same cdepth.

**Lemma B.2.** *Let $X \in \mathbb{N}$, $\mathcal{X} = [[\pm X]]$, $\mathcal{S} \in (\mathcal{X}^d \times \mathcal{X})^*$, $i \in [d]$, let $x_1^*, \ldots, x_{i-1}^* \in \mathbb{R}$, let $Q_{x_1^*, \ldots, x_{i-1}^*}$ be the function from Definition 4.2 and let $\tilde{x}_i \in \mathbb{R}$ be a decreasing point for $Q_{x_1^*, \ldots, x_{i-1}^*}(\mathcal{S}, \cdot)$ (according to Definition B.1). Then there exists a subset $\mathcal{S}' \subseteq \mathcal{S}$ of size $\leq d - i + 1$ such that the set $\mathcal{H}_{\mathcal{S}'} \subseteq \mathbb{R}^{d-i+1}$ defined by $\mathcal{H}_{\mathcal{S}'} := \bigcap_{(\mathbf{a},w) \in \mathcal{S}'} \mathrm{hp}_{(a_i, \ldots, a_d), w - \sum_{j=1}^{i-1} a_j x_j^*}$ is not empty, and for every $(x_i, \ldots, x_d) \in \mathcal{H}_{\mathcal{S}'}$ it holds that $x_i = \tilde{x}_i$ and that $\mathrm{cdepth}_{\mathcal{S}}(x_1^*, \ldots, x_{i-1}^*, x_i, \ldots, x_d) = Q_{x_1^*, \ldots, x_{i-1}^*}(\mathcal{S}, \tilde{x}_i)$.*

*Proof.* We start by noting that for any $(\mathbf{a}, w) \in \mathcal{S}$, the hyperplane $\mathrm{hp}_{(a_i, \ldots, a_d), w - \sum_{j=1}^{i-1} a_j x_j^*}$ and the halfspace $\mathrm{hs}_{(a_i, \ldots, a_d), w - \sum_{j=1}^{i-1} a_j x_j^*}$ are simply the projections of the original hyperplane $\mathrm{hp}_{\mathbf{a}, w}$ and halfspace $\mathrm{hs}_{\mathbf{a}, w}$ (respectively) to the subspace $\mathcal{V} := \{\mathbf{x} \in \mathbb{R}^d \colon (x_1, \ldots, x_{i-1}) = (x_1^*, \ldots, x_{i-1}^*)\}$.[11] Therefore, in this projected $(d - i + 1)$-subspace, for each point $(x_i, \ldots, x_d) \in \mathbb{R}^{d-i+1}$ we have

$$\mathrm{depth}_{\mathcal{S}}(x_1^*, \ldots, x_{i-1}^*, x_i, \ldots, x_d) = \left| \left\{ (\mathbf{a}, w) \in \mathcal{S} \colon (x_i, \ldots, x_d) \in \mathrm{hs}_{(a_i, \ldots, a_d), w - \sum_{j=1}^{i-1} a_j x_j^*} \right\} \right|.$$

In the following, let $\tilde{x}_i$ be a decreasing point for $Q_{x_1^*, \ldots, x_{i-1}^*}(\mathcal{S}, \cdot)$, let $k = Q_{x_1^*, \ldots, x_{i-1}^*}(\mathcal{S}, \tilde{x}_i)$, and assume without loss of generality that for all $x_i < \tilde{x}_i$ it holds that $Q_{x_1^*, \ldots, x_{i-1}^*}(\mathcal{S}, x_i) < k$ (the case $x_i > \tilde{x}_i$ can be handled similarly). By definition of $Q_{x_1^*, \ldots, x_{i-1}^*}(\mathcal{S}, \cdot)$ and by the assumption on $\tilde{x}_i$, there exist $\tilde{x}_{i+1}, \ldots, \tilde{x}_d \in \mathbb{R}$ such that

$$
\begin{aligned}
k &= \mathrm{cdepth}_{\mathcal{S}}(x_1^*, \ldots, x_{i-1}^*, \tilde{x}_i, \ldots, \tilde{x}_d) \\
&= \mathrm{depth}_{\mathcal{S}}(x_1^*, \ldots, x_{i-1}^*, \tilde{x}_i, \ldots, \tilde{x}_d) \\
&= \left| \left\{ (\mathbf{a}, w) \in \mathcal{S} \colon (\tilde{x}_i, \ldots, \tilde{x}_d) \in \mathrm{hs}_{(a_i, \ldots, a_d), w - \sum_{j=1}^{i-1} a_j x_j^*} \right\} \right|.
\end{aligned}
$$

For justifying the second equality, note that $\mathrm{cdepth}_{\mathcal{S}}(x_1^*, \ldots, x_{i-1}^*, \tilde{x}_i, \ldots, \tilde{x}_d) = k$ implies by definition that $(x_1^*, \ldots, x_{i-1}^*, \tilde{x}_i, \ldots, \tilde{x}_d)$ is a convex combination of points with $\mathrm{depth}_{\mathcal{S}} \geq k$. Since $\tilde{x}_i$ is a decreasing point, non of these points have $x_i < \tilde{x}_i$, and therefore all these points have $x_i = \tilde{x}_i$. This yields the existence of such $\tilde{x}_{i+1}, \ldots, \tilde{x}_d$ with $\mathrm{depth}_{\mathcal{S}}(x_1^*, \ldots, x_{i-1}^*, \tilde{x}_i, \ldots, \tilde{x}_d) = k$.

By the above equation, for every $x_i, \ldots, x_d \in \mathbb{R}$ with $x_i < \tilde{x}_i$ it holds that

$$\mathrm{depth}_{\mathcal{S}}(x_1^*, \ldots, x_{i-1}^*, x_i, \ldots, x_d) \leq Q_{x_1^*, \ldots, x_{i-1}^*}(\mathcal{S}, x_i) < k = \mathrm{depth}_{\mathcal{S}}(x_1^*, \ldots, x_{i-1}^*, \tilde{x}_i, \ldots, \tilde{x}_d) \qquad (4)$$

We now construct the set $\mathcal{S}'$. We initialize it to

$$\mathcal{S}' := \left\{ (\mathbf{a}, w) \in \mathcal{S} \colon (\tilde{x}_i, \ldots, \tilde{x}_d) \in \mathrm{hp}_{(a_i, \ldots, a_d), w - \sum_{j=1}^{i-1} a_j x_j^*} \right\}$$

Note that $\mathcal{H}_{\mathcal{S}'}$ is a hyperplane of dimension $\leq d - i$ (possibly the 1-dimensional hyperplane which is just the single point $\{(\tilde{x}_i, \ldots, \tilde{x}_d)\}$). Assume towards a contradiction that $\mathcal{H}_{\mathcal{S}'}$ contains a point $(x_i', \ldots, x_d')$ with $x_i' \neq \tilde{x}_i$. Then $\mathcal{H}_{\mathcal{S}'}$ must be a hyperplane of dimension at least two that in particular contains the line (in $\mathbb{R}^{d-i+1}$) that is determined by $(\tilde{x}_i, \ldots, \tilde{x}_d)$ and $(x_i', \ldots, x_d')$. In particular, this line contains a point $(x_i'', \ldots, x_d'')$ with $x_i'' < \tilde{x}_i$ such that the distance between $(\tilde{x}_i, \ldots, \tilde{x}_d)$ and $(x_i'', \ldots, x_d'')$ is $\gamma/2$, letting $\gamma > 0$ be a positive bound on the distance between $(\tilde{x}_i, \ldots, \tilde{x}_d)$ to all the hyperplanes $\mathrm{hp}_{(a_i, \ldots, a_d), w - \sum_{j=1}^{i-1} a_j x_j^*}$ for $(\mathbf{a}, w) \in \mathcal{S} \setminus \mathcal{S}'$ (i.e., hyperplanes that $(\tilde{x}_i, \ldots, \tilde{x}_d)$ does not lie on them). It is easy to verify that $(x_i'', \ldots, x_d'')$ belongs to exactly the same (projected) halfspaces that $(\tilde{x}_i, \ldots, \tilde{x}_d)$ do: Both belong to the intersection

of hyperplanes that are defined by $\mathcal{S}'$ (i.e., $\mathcal{H}_{\mathcal{S}'}$), and belong to the same side of the hyperplanes defined by $\mathcal{S} \setminus \mathcal{S}'$. Therefore $\text{depth}_{\mathcal{S}}(x_1^*, \ldots, x_{i-1}^*, x_i'', \ldots, x_d'') = \text{depth}_{\mathcal{S}}(x_1^*, \ldots, x_{i-1}^*, \tilde{x}_i, \ldots, \tilde{x}_d)$, in contradiction to Equation (4).

At this point, we constructed $\mathcal{S}'$ which is not empty, and all the points in $\mathcal{H}_{\mathcal{S}'}$ have $x_i = \tilde{x}_i$, and there is at least one point $(\tilde{x}_i, x_{i+1}, \ldots, x_d) \in \mathcal{H}_{\mathcal{S}'}$ with $\text{cdepth}_{\mathcal{S}}(x_1^*, \ldots, x_{i-1}^*, \tilde{x}_i, x_{i+1}, \ldots, x_d) = k$. If all the points in $\mathcal{H}_{\mathcal{S}'}$ reaches $k$, then we are done. Otherwise, define

$$Q'_{x_1^*, \ldots, x_{i-1}^*, \tilde{x}_i}(\mathcal{S}, x_{i+1}) := \max_{\substack{(x_{i+2}, \ldots, x_d): \\ (\tilde{x}_i, x_{i+1}, x_{i+2} \ldots, x_d) \in \mathcal{H}_{\mathcal{S}'}}} \text{cdepth}_{\mathcal{S}}(x_1^*, \ldots, x_{i-1}^*, \tilde{x}_i, x_{i+1}, \ldots, x_d)$$

If this function is constant, then fix an arbitrary $\tilde{x}_{i+1} \in \mathbb{R}$ for the next iteration. Otherwise, this function must have a decreasing point $\tilde{x}_{i+1}$ with $Q'_{x_1^*, \ldots, x_{i-1}^*, \tilde{x}_i}(\mathcal{S}, \tilde{x}_{i+1}) = k$. By the same arguments done before, we can add more pairs to $\mathcal{S}'$ such that now all the points in $\mathcal{H}_{\mathcal{S}'}$ have also $x_{i+1} = \tilde{x}_{i+1}$ and still there is at least one point that reaches cdepth of $k$. In both cases, for the next iteration, one can consider now the function

$$Q'_{x_1^*, \ldots, x_{i-1}^*, \tilde{x}_i, \tilde{x}_{i+1}}(\mathcal{S}, x_{i+2}) := \max_{\substack{(x_{i+3}, \ldots, x_d): \\ (\tilde{x}_i, \tilde{x}_{i+1}, x_{i+2}, \ldots, x_d) \in \mathcal{H}_{\mathcal{S}'}}} \text{cdepth}_{\mathcal{S}}(x_1^*, \ldots, x_{i-1}^*, \tilde{x}_i, \tilde{x}_{i+1}, x_{i+2}, \ldots, x_d),$$

for determine a value $\tilde{x}_{i+2}$, and so forth. Eventually, this process must end after at most $d - i + 1$ iteration, in which the resulting $\mathcal{H}_{\mathcal{S}'}$ satisfies that all the points that belongs to it reaches cdepth of $k$, as required.

At the end of the process, in case there are more than $d - i + 1$ hyperplanes in $\mathcal{S}'$, then there exists at least one hyperplane which is linearly depended in the others (i.e., its coefficients vector is linearly dependent in the coefficients vectors of the other hyperplanes in $\mathcal{S}'$). Therefore, by removing it from $\mathcal{S}'$ it does not change the intersection $\mathcal{H}_{\mathcal{S}'}$. Therefore, it is possible to remove hyperplanes from $\mathcal{S}'$ until $|\mathcal{S}'| = d - i + 1$. □

We now ready for proving Lemma 4.3, restated below.

**Lemma B.3** (Restatement of Lemma 4.3). *Let* $X \in \mathbb{N}$, $\mathcal{X} = [[\pm X]]$, $\mathcal{S} \in (\mathcal{X}^d \times \mathcal{X})^*$, $i \in [d]$, *let* $x_1^*, \ldots, x_{i-1}^* \in \mathbb{R}$ *and let* $Q_{x_1^*, \ldots, x_{i-1}^*}$ *be the function from Definition 4.2. Then there exists an invertible matrix* $\mathbf{A} \in \mathcal{X}^{(d-i+1) \times (d-i+1)}$ *and values*

$$b_i, \ldots, b_d \in \mathcal{X} - \sum_{j=1}^{i-1} x_j^* \cdot \mathcal{X} := \bigcup_{w, a_1, \ldots, a_{i-1} \in \mathcal{X}} \left\{ w - \sum_{j=1}^{i-1} a_j x_j^* \right\}$$

*such that* $(x_i^*, \ldots, x_d^*)^T := \mathbf{A}^{-1} \cdot (b_i, \ldots, b_d)^T$ *satisfies*

$$\text{cdepth}_{\mathcal{S}}(x_1^*, \ldots, x_d^*) = Q_{x_1^*, \ldots, x_{i-1}^*}(\mathcal{S}, x_i^*) = \max_{x_i \in \mathbb{R}} \left\{ Q_{x_1^*, \ldots, x_{i-1}^*}(\mathcal{S}, x_i) \right\}.$$

*Proof.* Let $k = \max_{x_i \in \mathbb{R}} \left\{ Q_{x_1^*, \ldots, x_{i-1}^*}(\mathcal{S}, x_i) \right\}$. If the function $Q_{x_1^*, \ldots, x_{i-1}^*}(\mathcal{S}, \cdot)$ is constant, then the proof trivially follows. Otherwise, since the set $\mathcal{C}_{\mathcal{S}}(k)$ is closed, there must exists a decreasing point $x_i^*$ for $Q_{x_1^*, \ldots, x_{i-1}^*}(\mathcal{S}, \cdot)$ with $Q_{x_1^*, \ldots, x_{i-1}^*}(\mathcal{S}, x_i^*) = k$. By Lemma B.2, there exists a subset $\mathcal{S}' \subseteq \mathcal{S}$ such that the set $\mathcal{H}_{\mathcal{S}'} \subseteq \mathbb{R}^{d-i+1}$ defined by $\mathcal{H}_{\mathcal{S}'} := \bigcap_{(\mathbf{a}, w) \in \mathcal{S}'} \text{hp}_{(a_i, \ldots, a_d), w - \sum_{j=1}^{i-1} a_j x_j^*}$ is not empty, and for all $(x_i, \ldots, x_d) \in \mathcal{H}_{\mathcal{S}'}$ it holds that $x_i = x_i^*$ and that $\text{cdepth}_{\mathcal{S}}(x_1^*, \ldots, x_{i-1}^*, x_i, \ldots, x_d) = k$. We can assume without loss of generality that the vectors $\{(a_i, \ldots, a_d): (\mathbf{a}, w) \in \mathcal{S}'\}$ are linearly independent (otherwise, one can remove pairs from $\mathcal{S}'$ without changing $\mathcal{H}_{\mathcal{S}'}$). If $|\mathcal{S}'| = d - i + 1$ then we are done by defining the matrix $\mathbf{A}$ to be the matrix with rows $\{(a_i, \ldots, a_d): (\mathbf{a}, w) \in \mathcal{S}'\}$. Otherwise, one can add linearly independent rows from the grid (e.g., unit vectors) without changing the properties of $\mathcal{H}_{\mathcal{S}'}$. The proof now follows. □

## B.2 Implementing $\mathcal{A}_{\text{FindDeepPoint}}$

In this section we show how $\mathcal{A}_{\text{FindDeepPoint}}$ (Figure 2) can be implemented, and we bound its running time. The formal statement appears below.

**Lemma B.4.** *Let $\alpha, \beta, \varepsilon \leq 1$, $\delta < 1/2$, $X \in \mathbb{N}$, $\mathcal{X} = [[\pm X]]$ and let $\mathcal{S} \in \left(\mathcal{X}^d \times \mathcal{X}\right)^*$. Then $\mathcal{A}_{\mathrm{FindDeepPoint}}$ on input $\alpha, \beta, \varepsilon, \delta, \mathcal{S}$ runs in time*

$$T = \mathrm{poly}(d) \cdot |\mathcal{S}| \cdot \left(|\mathcal{S}|^d \cdot \log X + \mathrm{polylog}(1/\alpha, 1/\beta, 1/\varepsilon, 1/\delta, X)\right).$$

*Proof.* We show how to implement in time $\mathrm{poly}(d) \cdot |\mathcal{S}| \cdot \left(|\mathcal{S}|^d \cdot \log X + \mathrm{polylog}(1/\alpha, 1/\beta, 1/\varepsilon, 1/\delta, X)\right)$ each iteration $i \in [d]$ of $\mathcal{A}_{\mathrm{OptimizeHighDimFunc}}$ (Figure 1). At the beginning of the iteration, we first start with a preprocessing phase that takes time $|\mathcal{S}|^{d+1} \cdot \mathrm{poly}(d) \cdot \log X$ in which we construct a list $L$ of size $O(|\mathcal{S}| \cdot \log |\tilde{\mathcal{X}}_i|) \leq O(d^2 \log d \cdot |\mathcal{S}| \cdot \log X)$. This list contains all pairs $(x_i^*, k) \in \tilde{\mathcal{X}}_i \times [[\mathcal{S}]]$ (in sorted order according to the first value) such that $k = Q_{x_1^*,\ldots,x_{i-1}^*}(\mathcal{S}, x_i^*)$ and $x_i^*$ is a decreasing point for $Q_{x_1^*,\ldots,x_{i-1}^*}(\mathcal{S}, \cdot)$ according to Definition B.1. Furthermore, the list also contain $(-\tilde{X}_i, Q_{x_1^*,\ldots,x_{i-1}^*}(\mathcal{S}, -\tilde{X}_i))$ and $(\tilde{X}_i, Q_{x_1^*,\ldots,x_{i-1}^*}(\mathcal{S}, \tilde{X}_i))$, letting $\tilde{X}_i = \max(\tilde{\mathcal{X}}_i)$. In order to compute $Q_{x_1^*,\ldots,x_{i-1}^*}(\mathcal{S}, x_i)$ for some $x_i \in \tilde{\mathcal{X}}_i$, we search in the list two adjacent pairs $(x_i', k')$ and $(x_i'', k'')$ such that $x_i \in [x_i', x_i'']$, and then it just holds that $Q_{x_1^*,\ldots,x_{i-1}^*}(\mathcal{S}, x_i) = \min\{k', k''\}$ (the direction $\geq$ is clear since the function is quasi-concave. For the other direction, note that if $Q_{x_1^*,\ldots,x_{i-1}^*}(\mathcal{S}, x_i) > \min\{k', k''\}$, where assume without loss of generality that $k' \leq k$, then there must exists a decreasing point between $x_i'$ and $x_i$ since the sets $\mathcal{C}_\mathcal{S}(\cdot)$ are close, in contradiction to the assumption that $L$ contains all decreasing points). This computation can be done in time $O(|\mathcal{S}| \cdot \tilde{Y}_i)$, letting $\tilde{Y}_i = \tilde{O}(d^4 \log X)$ be the number of bits that are needed for representing all the points in $\tilde{\mathcal{X}}_i$. Similarly, given $j \in [\tilde{Y}_i]$, computing $L(\mathcal{S}, j)$ can be performed by searching pairs $(x_i', k')$ and $(x_i'', k'')$ with $x_i'' - x_i' \geq 2^j$ that maximize $\min\{k', k''\}$. This can also be implement in time $O(|\mathcal{S}| \cdot \tilde{Y}_i)$. Therefore, given the list $L$, we conclude by the above analysis along with Fact A.1 that $\mathcal{A}_{\mathrm{RecConcave}}$ can be implemented in time $\mathrm{poly}(d) \cdot |\mathcal{S}| \cdot \mathrm{polylog}(1/\alpha, 1/\beta, 1/\varepsilon, 1/\delta, X)$.

The expensive part is constructing the list $L$. By Lemma B.2, in order to find all decreasing points with their values, it is enough to go over all the $O(|\mathcal{S}|^{d-i+1})$ intersections between at most $d - i + 1$ hyperplane in the set $\bigcup_{(\mathbf{a}, w) \in \mathcal{S}} \left\{ \mathrm{hp}_{(a_i,\ldots,a_d), w - \sum_{j=1}^{i-1} a_j x_j^*} \right\}$ and check whether they uniquely determine that $x_i = \tilde{x}_i$ for some $\tilde{x}_i \in \mathbb{R}$. For each such $\tilde{x}_i$, find $\tilde{x_{i+1}}, \ldots, \tilde{x}_d \in \mathbb{R}$ such that $(\tilde{x}_i, \ldots, \tilde{x}_d)$ belongs to the intersection, evaluate the depth $k = \mathrm{depth}_\mathcal{S}(x_1^*, \ldots, x_{i-1}^*, \tilde{x}_i, \ldots, \tilde{x_d})$ and update the list: if there exists $(x_i', k'), (x_i'', k'')$ in $L$ such that $\tilde{x}_i \in [x_i', x_i'']$ and $k \leq \min\{k', k''\}$, then ignore $\tilde{x}_i$ (it is not a decreasing point). Otherwise, insert $\tilde{x}_i$ to the list and remove all points $(x_i', k')$ that we know they are not a decreasing point after this insertion. Checking whether the intersection uniquely determine $x_i$ and finding a point in it, can be done in time $\mathrm{poly}(d) \log \tilde{X}_i$ using Guassian elimination. Since the size of the list is $O(|\mathcal{S}|)$ in each step, updating the list each time can be done in time $O(|\mathcal{S}| \log \tilde{X}_i)$. $\qquad\square$

## B.3 Proving Theorem 5.2

In this section we present the proof of Theorem 5.2. We start by stating two lemmatas. The first lemma states that if the points in the dataset are coming from a grid $\mathcal{X}^d = [[\pm X]]^d$, then there is a margin of $1/(d \cdot X)^{\mathrm{poly}(d)}$.

**Lemma B.5.** *Let $X \in \mathbb{N}$, $\mathcal{X} = [[\pm X]]$ and let $\mathcal{S} \in \left(\mathcal{X}^d \times \{-1, 1\}\right)^*$ be a realizable dataset of points. Then there exists a halfspace $\mathrm{hs} \subset \mathbb{R}^d$ with $\mathrm{val}_\mathcal{S}(\mathrm{hs}) = |\mathcal{S}|$ such that for all $(\mathbf{x}, \cdot) \in \mathcal{S}$ it holds that $\mathrm{dist}(\mathbf{x}, \mathrm{hs}) := \min_{\mathbf{x}' \in \mathrm{hs}} \{\|\mathbf{x} - \mathbf{x}'\|\} \geq 1/X'$, for $X' := 2d^2 \cdot d!^{d^3} \cdot X^{d^6}$.*

*Proof.* We prove that $\exists \mathbf{a} = (a_1, \ldots, a_d) \in \mathbb{R}^d$ with $a_i \in [[\pm d!^d \cdot X^{d^2}]] / \left(d \cdot [[\pm d!^d \cdot X^{d^2}]] \setminus \{0\}\right)$ and $w \in \{-1, 0, 1\}$ such that $\mathrm{val}_\mathcal{S}(\mathrm{hs}_{\mathbf{a}, w}) = |\mathcal{S}|$. This yields that for any $\mathbf{x} \in \mathcal{X}$ we have that

$$\langle \mathbf{a}, \mathbf{x} \rangle \in [[\pm d!^{d^2} \cdot X^{d^4}]] / \left(d \cdot [[\pm d!^{d^2} \cdot X^{d^4}]] \setminus \{0\}\right).$$

Therefore, for every $(\mathbf{x}, -1) \in \mathcal{S}$, since $\langle \mathbf{a}, \mathbf{x} \rangle < w$ then it must hold that $\langle \mathbf{a}, \mathbf{x} \rangle \leq w - 1/\left(d \cdot d!^{d^2} \cdot X^{d^4}\right)$. This yields that for every $(\mathbf{x}, -1) \in \mathcal{S}$ and every $\mathbf{v} \in \mathbb{R}^d$ with $\|\mathbf{v}\| < 2/X'$ it holds that

$$\langle \mathbf{a}, \mathbf{x} + \mathbf{v} \rangle \leq \langle \mathbf{a}, \mathbf{x} \rangle + \|\mathbf{a}\| \cdot \|\mathbf{v}\| \leq \left(w - 1/\left(d \cdot d!^{d^2} \cdot X^{d^4}\right)\right) + \left(d \cdot d!^d \cdot X^{d^2}\right) \cdot 2/X' < w.$$

At this point we proved the existence of a halfspace hs with $\text{val}_{\mathcal{S}}(\text{hs}) = |\mathcal{S}|$ such that it is far by at least $2/X'$ from all the point $\mathbf{x}$ with $(\mathbf{x}, -1) \in \mathcal{S}$. This in particular yields the existence of an halfspace hs$'$ with $\text{val}_{\mathcal{S}}(\text{hs}') = |\mathcal{S}|$ that is far by at least $1/X'$ from all the points in $\mathcal{S}$.

It remains to prove the existence of such $\mathbf{a}$ and $w$. As explained in Section 5, the assumption that $\mathcal{S}$ is a realizable dataset of points implies that there exists $w \in \{-1, 0, 1\}$ such that there exists a solution $\mathbf{a} = (a_1, \ldots, a_d) \in \mathbb{R}^d$ to the system of equations

$$\mathcal{E} := \{\langle \mathbf{x}, \mathbf{a} \rangle \geq w\}_{(\mathbf{x},1) \in \mathcal{S}} \bigcup \{\langle -\mathbf{x}, \mathbf{a} \rangle > -w\}_{(\mathbf{x},-1) \in \mathcal{S}} \, .$$

Let $\mathcal{F}$ be the feasible area of $\mathcal{E}$, and let $C(\mathcal{F})$ be the closure of $\mathcal{F}$ which is a polytope in $\mathbb{R}^d$ (might be unbounded). Each vertex of $C(\mathcal{F})$ is a solution to $d$ linearly independent equations in $\{\langle y \cdot \mathbf{x}, \mathbf{a} \rangle = y \cdot w\}_{(\mathbf{x},y) \in \mathcal{S}}$. Therefore, for any vertex $\mathbf{a}^* = (a_1^*, \ldots, a_d^*)$, it holds by Cramer's rule that $a_i^* \in [[\pm d! \cdot X^{d-1}]] / ([[\pm d! \cdot X^d]] \setminus \{0\})$. Let $d' \leq d$ be the (largest) value in which $C(\mathcal{F})$ has $d'$-dimensional non-zero volume. If $C(\mathcal{F})$ has less than $d' + 1$ vertices, then $C(\mathcal{F})$ is unbounded and the statement trivially follows. Otherwise, the average of $d' + 1$ vertices of $C(\mathcal{F})$ must be a point in $\mathcal{F}$ and the proof follows since each coordinate of the average belongs to

$$\sum_{j=1}^{d'} [[\pm d! \cdot X^{d-1}]] / \left( d \cdot [[\pm d! \cdot X^d]] \setminus \{0\} \right) \subseteq [[\pm d!^d \cdot X^{d^2}]] / \left( d \cdot [[\pm d!^d \cdot X^{d^2}]] \setminus \{0\} \right)$$

$\square$

The second lemma determines the resolution of the noise that we need to add to each of the points in $\mathcal{S}$ in order to guarantee general position with high probability.

**Lemma B.6.** *Let $\mathcal{S} \subseteq (\mathbb{R}^d)^*$ be a multiset, let $\beta > 0$, and let $U_{\mathcal{A}}$ be the uniform distribution over a set $\mathcal{A} \subset \mathbb{R}$ of size $\geq d |\mathcal{S}|^d / \beta$. Let $\tilde{\mathcal{S}}$ be the multiset that is generated by the following process: For each $\mathbf{x} = (x_1, \ldots, x_d) \in \mathcal{S}$, sample $\mathbf{z} = (z_1, \ldots, z_d) \sim (U_{\mathcal{A}})^d$ (i.e., each $z_i$ is sampled independently from $U_{\mathcal{A}}$), and insert $(x_1 + z_1, \ldots, x_d + z_d)$ to $\tilde{\mathcal{S}}$. Then with probability at least $1 - \beta$ it holds that the points in $\tilde{\mathcal{S}}$ are in general position.*

*Proof.* Note that a set of points $\mathcal{S} \subset \mathbb{R}^d$ are in general position if for any $d+1$ points $\tilde{\mathbf{x}}_1 = (\tilde{x}_{1,1}, \ldots, \tilde{x}_{1,d}), \ldots, \tilde{\mathbf{x}}_{d+1} = (\tilde{x}_{d+1,1}, \ldots, \tilde{x}_{d+1,d}) \in \tilde{\mathcal{S}}$ it holds that the vectors $(\tilde{\mathbf{x}}_1 - \tilde{\mathbf{x}}_{d+1}), \ldots, (\tilde{\mathbf{x}}_d - \tilde{\mathbf{x}}_{d+1})$ are linearly independent, meaning that $\det\left((\tilde{x}_{i,j} - \tilde{x}_{d+1,j})_{i,j \in [d]}\right) \neq 0$. In the following, for $k \in [d]$, let $E_k$ be the event that for all $k$ points $\tilde{\mathbf{x}}_1, \ldots, \tilde{\mathbf{x}}_{k-1}, \tilde{\mathbf{x}}_{d+1} \in \tilde{\mathcal{S}}$ it holds that the $k \times k$ matrix $(\tilde{x}_{i,j} - \tilde{x}_{d+1,j})_{i,j \in [k]}$ has determinant $\neq 0$. Our goal is to show that $\Pr[E_d] \geq 1 - \beta$, which yields that the points in $\tilde{\mathcal{S}}$ are in general position w.p. $\geq 1 - \beta$. We start with the event $E_1$. The event means that all the points in $\tilde{\mathcal{S}}$ has first coordinate $\neq 0$. Since the first coordinate is taken uniformly from a set of size $|\mathcal{A}|$, then by union bound the probability that one of the points has first coordinate $0$ is bounded by $|\mathcal{S}| / |\mathcal{A}|$, meaning that $\Pr[\neg E_1] \leq |\mathcal{S}| / |\mathcal{A}|$. We now prove that for each $k \in [d]$ it holds that $\Pr[\neg E_k \mid E_1 \wedge \ldots \wedge E_{k-1}] \leq |\mathcal{S}|^k / |\mathcal{A}|$. Fix $k$ points $\tilde{\mathbf{x}}_1, \ldots, \tilde{\mathbf{x}}_{k-1}, \tilde{\mathbf{x}}_{d+1} \in \tilde{\mathcal{S}}$. Note that by computing the determinant of $(\tilde{x}_{i,j} - \tilde{x}_{d+1,j})_{i,j \in [k]}$ using its last row we get that $\det((\tilde{x}_{i,j} - \tilde{x}_{d+1,j})_{i,j \in [k]}) = (-1)^k \cdot \det\left((\tilde{x}_{i,j} - \tilde{x}_{d+1,j})_{i,j \in [k-1]}\right) \cdot (\tilde{x}_{k,k} - \tilde{x}_{d+1,k}) + \lambda$, where $\lambda$ is independent of $\tilde{x}_{k,k}$, and $\det\left((\tilde{x}_{i,j} - \tilde{x}_{d+1,j})_{i,j \in [k-1]}\right) \neq 0$ by the conditioning. Therefore, in order for the determinant to be $0$, it must hold that $\tilde{x}_{k,k} = \tilde{x}_{d+1,k} + (-1)^{k+1} \cdot \lambda / \det\left((\tilde{x}_{i,j})_{i,j \in [k-1]}\right)$. This holds with probability at most $1/|\mathcal{A}|$ for any such fixing of $k$ points, and therefore we deduce by union bound that $\Pr[\neg E_k \mid E_1 \wedge \ldots \wedge E_{k-1}] \leq |\mathcal{S}|^k / |\mathcal{A}|$. We conclude that

$$\Pr[E_d] \geq \Pr[E_1 \wedge \ldots \wedge E_d] = 1 - \sum_{k=1}^{d} \Pr[\neg E_k \mid E_1 \wedge \ldots \wedge E_{k-1}] \geq 1 - d \cdot |\mathcal{S}|^d / |\mathcal{A}| \geq 1 - \beta$$

$\square$

We now ready to prove Theorem 5.2, restated below.

**Theorem B.7** (Restatement of Theorem 5.2). *Let $\alpha, \beta, \varepsilon \leq 1$, $\delta < 1/2$, $X \in \mathbb{N}$ and let $\mathcal{X} = [[\pm X]]$. Then there exists an $(\varepsilon, \delta)$-differentially private $(\alpha, \beta)$-PAC learner with sample complexity $s$ for the class* $\mathtt{HALFSPACE}(\mathcal{X}^d)$ *for* $s = O\left(d^{2.5} \cdot 2^{O\left(\log^* X + \log^* d + \log^*\left(\frac{1}{\alpha\beta\varepsilon\delta}\right)\right)} \cdot \frac{\log^{1.5}\left(\frac{1}{\delta}\right)\log\left(\frac{d}{\alpha\beta}\right)}{\varepsilon\alpha}\right)$.

*Proof.* Let $\mu$ be a target distribution over points in $\mathcal{X}^d$. In the following, let $X'$ be the value from Lemma B.5, let $\Delta := \lceil d \cdot s^d/(2\beta) \rceil$, let $\Delta' := 2\Delta \cdot X'\sqrt{d}$ and let $\mathcal{A} := [[\pm\Delta]]/\Delta'$. We now define the (noisy) distribution $\tilde{\mu} := \mu + (U_{\mathcal{A}})^d$ (Namely, $\tilde{\mu}$ is the distribution induces by the outcome of $\mathbf{x} + \mathbf{z}$ where $\mathbf{x} \sim \mu$ and $\mathbf{z} \sim (U_{\mathcal{A}})^d$, i.e., each $z_i$ is sampled independently and uniformly from $\mathcal{A}$). Note that $\tilde{\mu}$ can be seen as a distribution over points in $\tilde{\mathcal{X}}^d = [[\pm\tilde{X}]]^d$, for $\tilde{X} := \Delta'(X + \Delta)$ (one just need to strech the points from $\mu$ by a factor of $\Delta'$ in order to guarantee that they will be on an integer grid).

Consider now an $s$-size dataset $\mathcal{S} \in (\mathcal{X}^d \times \{-1, 1\})$ where the points in $\mathcal{S}$ are sampled according to $\mu$ and the labels are according to a concept function $c \in \mathtt{HALFSPACE}(\mathcal{X}^d)$. We now construct a dataset $\mathcal{S}' \in (\tilde{\mathcal{X}}^d \times \{-1, 1\})$, where for each $(\mathbf{x}, y) \in \mathcal{S}$ we insert $(\mathbf{x}+\mathbf{z}, y)$ into $\mathcal{S}'$, for a random noise $\mathbf{z} \sim (U_{\mathcal{A}})^d$. Since, by definition, it holds that $\|z\| < 1/X'$, then by Lemma B.5 we deduce that the dataset $\mathcal{S}'$ remains realizable. By Lemma B.6, since $|\mathcal{A}| \geq d |\mathcal{S}|^d/(4\beta)$, it holds that the points in $\tilde{\mathcal{S}}$ are in general position (except with probability $\beta/4$). Therefore, by the above arguments and by Theorem 5.1, when executing $\mathcal{A}_{\text{LearnHalfSpace}}$ on the dataset $\mathcal{S}'$ and the parameters $\alpha/20, \beta/4, \varepsilon, \delta$, then with probability $\geq 1 - \beta/2$ the resulting hypothesis $h = c_{\mathbf{a},w}$ satisfies that $h(\mathbf{x}) = y$ for at least $(1 - \alpha/20)|\tilde{\mathcal{S}}|$ of the pairs $(\mathbf{x}, y) \in \tilde{\mathcal{S}}$, where recall that $c_{\mathbf{a},w}(\mathbf{x}) = 1 \iff \mathbf{x} \in \text{hs}_{\mathbf{a},w}$. By Theorem 2.8, we decude that $\Pr_{h \sim \mathcal{A}_{\text{LearnHalfSpace}}}[\text{error}_{\tilde{\mu}}(c, h) \leq \alpha/2] \geq 1 - \beta$. We finish the proof by showing that for every $h$ it holds that $\text{error}_{\mu}(c, h) \leq 2 \cdot \text{error}_{\tilde{\mu}}(c, h)$. For that, note that

$$\text{error}_{\tilde{\mu}}(c, h) = \Pr_{\mathbf{x}+\mathbf{z}\sim\tilde{\mu}}[c(\mathbf{x} + \mathbf{z}) \neq h(\mathbf{x} + \mathbf{z})]$$
$$\geq \Pr_{\mathbf{x}\sim\mu}[c(\mathbf{x}) \neq h(\mathbf{x})] \cdot \Pr_{\mathbf{x}+\mathbf{z}\sim\tilde{\mu}}[c(\mathbf{x} + \mathbf{z}) \neq h(\mathbf{x} + \mathbf{z}) \mid c(\mathbf{x}) \neq h(\mathbf{x})]$$

Hence, it is enough to show that for every $\mathbf{x} \in \mathbb{R}^d$ such that $c(\mathbf{x}) \neq h(\mathbf{x})$ it holds that $c(\mathbf{x} + \mathbf{z}) \neq h(\mathbf{x} + \mathbf{z})$ with probability at least $1/2$. Assume without loss of generality that $h(\mathbf{x}) = 1$ and $c(\mathbf{x}) = -1$ (the other case can be handled similarly). The assumption $h(\mathbf{x}) = 1$ implies that $\langle \mathbf{a}, \mathbf{x} \rangle \geq w$ for the $\mathbf{a}, w$ that $h = c_{\mathbf{a},w}$. Note that for all $\mathbf{z} \in \mathcal{A}^d$ it holds that at least one of $\{\mathbf{z}, -\mathbf{z}\}$ satisfies $\langle \mathbf{a}, \mathbf{z} \rangle \geq 0$ which implies that $\langle \mathbf{a}, \mathbf{x} + \mathbf{z} \rangle \geq w$. We deduce that at least half of the points in $\mathcal{A}^d$ satisfies $h(\mathbf{x} + \mathbf{z}) = h(\mathbf{x})$. The proof now follows since z is chosen uniformly from $\mathcal{A}^d$. $\qquad\square$