[Reviews · NeurIPS 2020]

Review 1

Summary and Contributions: This paper considers the problem of privately learning halfspaces over a discrete domain X. It is known that privately learning halfspaces requires sample complexity that scales with log*(|X|) (so learning over an infinite domain is impossible), and existing algorithms have dependence poly(d) * 2^log*|X|. This paper gives a simpler construction that improves the dependence on d from d^(4.5) to d^(2.5).

Strengths: The paper gives a clear reduction from the problem of learning halfspaces to the more general problem of finding solutions to linear programs that satisfy -most- of the constraints (there is one constraint for every data point --- that it lie on the correct side of the learned hyperplane). This problem is potentially more generally useful; it is solved by a reduction to the problem of maximizing a quasi-concave function, which is solvable by algorithms from prior work.

Weaknesses: The algorithm in this paper is in the end not self contained: it relies on an algorithm from prior work that maximizes quasi-concave functions. As a result, it is difficult from reading this paper to get any sense of the algorithm (since the algorithm is ultimately not contained in this paper). The paper sells one of its contributions as "simplifying the construction" --- so it would be nice if it were self contained, so that a reader who started with this paper could actually understand what the algorithm does! I -strongly- recommend including a summary of how the quasi-concave optimization works in the body of the paper and including the pseudo-code in the appendix.

Correctness: Yes

Clarity: Yes

Relation to Prior Work: Yes

Reproducibility: Yes

Additional Feedback:


Review 2

Summary and Contributions: This work improves the state of the art for differentially (approximate) private learning halfspaces over a finite grid in R^d (note that even in 1 dimension, privately learning halfspaces is not possible over R). If the grid is made of natural numbers bounded by X, the sample complexity is ~d^{2.5} * log*X where log* is the iterative log. The previous best, due to Beimel et al. 2019 is ~d^{4.5} * log*X, and the generic bound from Kasiviswanathan et al. 2011 gives d^2*log X, i.e. these results are interesting when log X is large compared to d. The algorithm: The algorithm, like Beimel et al. 2019, is based on the Beimel et al. 2016 (approximate) private algorithm for approximately maximizing a one-dimensional quasi-concave function (here the function is Q(S, *) where S is a database and Q(*,x) has sensitivity 1). This algorithm guarantees an approximation when the maximum is not too small compared to the domain. In Beimel et al. 2019, the 1-d quasi-concave maximizer is applied iteratively to privately find, in the end, a point in the convex hull of the points in a database. At each step, they maximize the quasi-concave function that computes the largest Tukey depth of a point, over all points possible by completion of the unset variables. Finally, the task of learning halfspaces is reduced to the convex hull problem. In this work, halfspace learning is easily reduced to the Linear Feasibility problem. Similar to Beimel et al., the quasi-concave maximizer is iteratively applied; but here, the function being maximized is simply the number of constraints satisfied in the dataset (by completion of unset variables). This function is not quasi-concave, but the authors propose a general technique called convexification which transforms the objective into a quasi-concave one. Care must be taken to ensure that: 1. A solution can still be found privately at each step, and 2. A solution that approximately maximizes the new function will approximately maximize the original. In summary, this work makes a couple contributions: - It improves on state of the art sample complexity by d^2 factor for privately learning halfspaces on a d-dimensional grid. - It gives a simpler more direct algorithm and proof. - Gives a new generic technique for turning a recursive objective function into a quasi-concave one.

Strengths: This work substantially improves on the dependence of d in the sample complexity compared to the previous algorithm. It is also arguably more direct in the approach and proof and solves a new and natural problem along the way (linear feasibility). It also introduces a possibly interesting paradigm for maximizing high dimensional functions that are not quasi-concave (but can be approximated by one in the appropriate sense). Update based on author's response ============================ My view on the paper, which was already quite positive, did not change based on the author feedback. ============================

Weaknesses: It’s not clear how close this is to the ‘right’ answer for approximately privately learning halfspaces on the grid; as the authors mention, recent work removes the exponential dependence on log*X in 1-d, and this paper does not speculate a lower bound. The approach here is not drastically different from Beimel et al 2019, but there is certainly technical novelty here. For how this work may impact other researchers, it might be helpful if the authors were to offer some thoughts on whether a linear dependence on log*X is possible in higher dimensions and what any obstacles would be to achieving that. It also would be nice to offer some possible uses or obstacles to using the generalized QuasiConcave paradigm -- while proof of requirement 3.2 was simple for linear feasibility, it was more involved to define the domain at each iterative step.

Correctness: I checked most of the details in the main submission to the best of my abilities and results are correct.

Clarity: The paper is quite well written and proofs well explained.

Relation to Prior Work: This paper does a good job of explaining prior work on this problem. Since the techniques here are

Reproducibility: Yes

Additional Feedback: It seems the reference in line 55 should be Beimel 2016?


Review 3

Summary and Contributions: This paper proposes an approximate differentially private learner for half spaces over a finite grid with sample complexity of O(d^2.5), which has improved the state-of-the-art result by O(d^2). As a side result, it also proposes algorithms for the linear feasibility problem, or in general, optimization problem where convexification incurs a small loss.

Strengths: This paper considers the problem of differentially privately learning half spaces, which is both an interesting and fundamental problem. This paper gives a new algorithm which has improved the state-of-the-art result by O(d^2), which is non-trivial. As a side result, it also proposes solutions for the linear feasibility problem, which is another important problem. The algorithm itself is nice and clean, which has utilized the results from [Beimei et al., 2016].

Weaknesses: Just a small point, I think it will be more clear if the authors can define the problem of learning half spaces in the introduction or preliminary. The problem definition appears in section 5, which is a little bit late. Besides, the problem is a little bit different from the problem in my mind, where there is no randomness in data generation. In other words, the problem considered in this paper is an ERM problem rather than a statistical learning problem. I think it is worth mentioning that the results in this paper can be easily generalized into the statistical setting, with an excess error of O(d/alpha^2), which is dominated by the empirical error.

Correctness: Yes.

Clarity: Generally speaking, it is well written and easy to understand. However, some polishing will be appreciated: 1. As mentioned in weakness, it will be better if the problem definition shows up in the intro or preliminary. 2. The symbols are a little messy. Please be consistent. 3. typo: page 2: "for which we know that there exist a completion", exist->exists 4. typo: at the end of page 5, it should be (1-alpha/delta) M_s instead of (1-alpha/delta) |S|.

Relation to Prior Work: Yes. However, more literature on DP learning theory will be appreciated, e.g., Dwork, Cynthia, and Vitaly Feldman. "Privacy-preserving prediction." arXiv preprint arXiv:1803.10266 (2018).

Reproducibility: Yes

Additional Feedback:

[Author Response · NeurIPS 2020]

We thank all three reviewers for their time and comments. Their suggestions will help us clarify the contributions of our
work as we incorporate them in the next revision of our paper.

In the following, we respond to several specific points raised by the reviewers.

Reviewer #1: You mentioned that the paper is at the end not self-contained because we didn't include a summary of
how the quasi-concave optimization (RecConcave) works. We take it into our attention and in the body of the final
version we will try to add a short summary of how it works, and to add more details in the appendix.

Reviewer #2: You wrote "it might be helpful if the authors were to offer some thoughts on whether a linear dependence
on $\log^* X$ is possible in higher dimensions". We take it into our attention. In the final version we will mention this
question, and write that one option for answering it is by finding a different 1-dimensional quasi-concave optimization
that is linear (or almost linear) in $\log^* X$, since RecConcave, the optimization that we are using, requires exponential
dependency in $\log^* X$. Indeed, a recent work of Kaplan, Ligett, Mansour, Naor, and Stemmer [COLT 2020] shows
an (almost) linear dependency in $\log^* X$ for 1-dimensional thresholds, which is a special case of a quasi-concave
optimization, and it still remains open whether this result can be extended to the quasi-concave optimization case.

Reviewer #2: You wrote that it would be nice to offer some possible uses or obstacles to using the generalized
QuasiConcave paradigm. Indeed, as you mentioned, for the linear feasibility it was more involved to define the domain
at each iterative step, and might be even more involved for other d-dimensional functions. The point is that this technical
issue is inherent for privately optimizing such functions (at least if the optimization is done coordinate by coordinate),
because we know that we must pay at least $\log^*$ of the domain size by any private algorithm. So we cannot get away
from finding finite domains. But, if we can find such domains with some finite bound on their sizes, even if it is very
large and not tight at all, it is usually should be enough since we are going to pay just $\log^*$ of these sizes in the sample
complexity. We will try to emphasize this point in the paper.

Reviewer #3: You wrote "it will be more clear if the authors can define the problem of learning half spaces in the
introduction or preliminary, ..." and "it is worth mentioning that the results in this paper can be easily generalized into
the statistical setting". We take all your suggestions into our attention. We will try to address them in the final version.

[Meta-Review · NeurIPS 2020]

There is a consensus among the reviewers that the paper provides new contribution to solving a fundamental problem of differentially private learning of half-spaces.